# Unsupervised Process-Aware Coreset Selection for In-Context Learning

**Wei Zheng** [1]  **Zijie Wang** [1]  **Xin Li** [1]  **Bin Gong** [1]  **Yuqing Sun** [1]

## Abstract

We address the challenge of unsupervised coreset selection for few-shot in-context learning (ICL). The goal is to select a small subset of examples under a fixed annotation budget to yield effective prompts for large language models. Existing geometry-based methods often yield coresets that suffer from a skewed distribution, due to the over-sampling of peripheral examples and high local redundancy. To address these issues, we propose a process-aware framework for coreset selection. It jointly optimizes the diversity and representativeness of selected samples via an adaptive sub-modular objective. It ensures representativeness by selecting samples based on local density awareness, while promoting diversity by imposing a redundancy penalty relative to the evolving selected set. Thus, it performs progress-aware balancing of representativeness and diversity based on the selection context. Extensive experiments on 7 NLP datasets demonstrate that our method consistently outperforms state-of-the-art coreset selection methods in downstream ICL performance. Further analysis validates that our approach better balances diversity and representativeness in the selection process, while retaining the theoretical guarantees of adaptive submodular optimization.

## 1. Introduction

In-Context Learning (ICL) enables large language models (LLMs) to perform new tasks by conditioning on a small set of input demonstrations. Empirically, ICL performance is highly sensitive to which demonstrations are used (Liu et al., 2022; Ye et al., 2023; Gao et al., 2024), especially when the available annotation budget is tight. It motivates the research on **Unsupervised Coreset Selection** (UCS) (Su et al., 2023; Zhang et al., 2024b): how to choose a small, informa-

tive subset from a large unlabeled pool so that the selected examples serve as effective in-context demonstrations.

Existing studies on UCS can be generally categorized into two lines. The first line comprises Model-based (or Active Learning) methods, which select examples by leveraging signals derived from a downstream model (Xie et al., 2023; Korakakis et al., 2024; Zhang et al., 2025a), such as predictive uncertainty, perplexity, or influence functions. However, these approaches incur prohibitive computational costs, as they typically require performing inference or iterative training over the unlabeled pool to estimate sample importance.

The second line consists of model-free methods, which select samples based on their geometric properties in the representation space. Prior work focuses on representativeness (Zhang et al., 2023; 2024a), aiming to select a coreset to cover the data distribution. However, strictly adhering to the density of the unlabeled pool often results in redundancy within the selected coreset. Since these methods lack awareness of the evolving selection process, they may repeatedly extract samples from the high-density regions. To mitigate this, recent studies have incorporated explicit diversity modeling (Shao et al., 2024; Tan et al., 2025; Su et al., 2023). Nevertheless, they often rely on static trade-offs, lacking a unified mechanism to dynamically calibrate the priority between diversity and representativeness across different budgets and the evolving selection process.

Consequently, existing methods have not established a unified framework that integrates the core requirements of unsupervised coreset selection: (i) achieving model- and label-agnostic, enabling direct operation on fully unlabeled pools; (ii) budget-adaptivity, capable of automatically calibrating the trade-off between representativeness and diversity under varying budgets (Zhang et al., 2024b); and (iii) process-aware representativeness, which prioritizes samples from high-density regions in the unlabeled pool while dynamically penalizing redundancy in the selection-process.

To address these gaps, we propose **Process-Aware Coreset Selection for In-Context Learning (PaCS-ICL)**, an unsupervised framework that jointly optimizes diversity and representativeness through a monotone adaptive submodular objective (Golovin & Krause, 2011). PaCS-ICL encourages diversity selection via a determinantal point process (DPP) (Kulesza & Taskar, 2012)-based term. We design a dy-

---

[1]School of Software, Shandong University, Jinan, China. Correspondence to: Yuqing Sun <sun_yuqing@sdu.edu.cn>.

*Proceedings of the $43^{rd}$ International Conference on Machine Learning*, Seoul, South Korea. PMLR 306, 2026. Copyright 2026 by the author(s).

namic representativeness term that modulates global density scores with a process-aware decay factor. This formulation ensures the selection prioritizes high-density regions in the pool, while preventing local overlaps by penalizing candidates located within the neighborhood radius of already selected samples. To ensure robustness across varying budgets, PaCS-ICL incorporates an adaptive balancing mechanism that calibrates the weights of the diversity and representativeness. By unifying these components, our approach admits an efficient greedy selection with theoretical approximation guarantees, establishing a robust and scalable solution for few-shot ICL.

We evaluate PaCS-ICL on 7 NLP datasets under tight annotation budgets. It consistently outperforms SOTA UCS methods in few-shot ICL. Ablation studies validate the effectiveness of the method design. Furthermore, PaCS-ICL proves to be computationally efficient, requiring neither iterative training nor human intervention.

## 2. Related Work

Prior works can be categorized into model-free and model-based methods. The most relevant to our work are model-free methods, which typically select coresets based on geometric properties of samples such as diversity and representativeness (Sener & Savarese, 2018; Zadeh et al., 2017; Feldman & Langberg, 2011; Zhang et al., 2023; 2024a). Recent SOTA approaches attempt to jointly evaluate these criteria using kernel-based measures or mutual information. For example, RDSS (Shao et al., 2024) uses kernel-based similarities and distributional consistency to evaluate diversity and representativeness, respectively. InfoMax (Tan et al., 2025) maximizes mutual information between the coreset and the unlabeled sample pool. However, balancing diversity and representativeness remains challenging (Zhang et al., 2024b).

The model-based methods utilize information obtained from the model training, such as difficulty scores (Xia et al., 2023; Yang et al., 2023) or gradient contributions (Paul et al., 2021), or leverage signals derived from downstream model inference, such as predictive uncertainty (Su et al., 2023; Xie et al., 2023). However, the model-derived signals result in prohibitive computational costs due to the need for iterative training or inference, limiting the practical utility. Unlike these methods, we focus on fully unlabeled datasets, without reliance on ground-truth labels or model feedback.

## 3. Method

### 3.1. Problem Setup and Framework

We consider the problem of Unlabeled Coreset Selection (UCS) for few-shot in-context learning (ICL). Let

$\mathcal{U} = \{x_1, x_2, \ldots, x_n\}$ denote a set of $n$ unlabeled samples. Given a budget $m$ (where $m \ll n$), the goal is to select a subset $S \subset \mathcal{U}$ of size $m$ for expert labeling (i.e., $|S| = m$) such that $S$ can form effective ICL prompts for unseen samples. Following prior works, we focus on extremely tight annotation budgets (e.g., $m \leq 100$), which are typical in few-shot ICL settings (Su et al., 2023; Zhang et al., 2024b).

Let $\text{Div}(S)$ denote the diversity of samples in $S$ and $\text{Rep}(S)$ measures how well $S$ represents the unlabeled samples in $\mathcal{U}$. The objective of UCS can then be formalized as maximizing a unified function $f(S)$, where $\alpha > 0$ is the parameter that controls the balance between diversity and representativeness. By adjusting $\alpha$, the method can flexibly adapt to different annotation budgets and data features. The function $f(S)$ thus jointly optimizes both criteria in selecting the coreset $S$.

$$\max_{S \subset \mathcal{U}, |S|=m} f(S),$$
$$f(S) = \text{Div}(S) + \alpha \text{Rep}(S) \tag{1}$$

**Framework.** We propose the Process-Aware Coreset Selection for In-Context Learning (PaCS-ICL), an unsupervised method that leverages submodular optimization—noted for its efficiency and guarantees—to select coresets by jointly maximizing diversity and representativeness through a monotone adaptive submodular objective.

In our design, $\text{Div}(S)$ is a standard monotone submodular function, while $\text{Rep}(S)$ is adaptively monotone and adaptively submodular. Consequently, their non-negative linear combination (e.g., Eq. 1) is also adaptively submodular due to the closure property. This allows us to efficiently maximize $f(S)$ using a greedy algorithm with a standard approximation guarantee. Namely, starting from an empty set $S_0 = \emptyset$, at each iteration $t$, we select a sample $x \in \mathcal{U}$ to form $S_t = S_{t-1} \cup \{x\}$ that provides the largest marginal gain, until $|S| = m$. Formally,

$$x^* = \arg \max_{x \in \mathcal{U} \setminus S_{t-1}} f(S_{t-1} \cup \{x\}) - f(S_{t-1}),$$
$$S_t = S_{t-1} \cup \{x^*\} \tag{2}$$

This algorithm achieves at least a $(1 - 1/e)$ fraction of the optimal objective (Nemhauser et al., 1978; Golovin & Krause, 2011), i.e., $f(S_{\text{greedy}}) \geq \left(1 - \frac{1}{e}\right) f(S^*)$, where $S^*$ is the optimal coreset of size $m$.

### 3.2. Submodular Diversity Term

According to the Determinantal Point Process (DPP) (Kulesza & Taskar, 2012), the probability of selecting a subset $S$ is proportional to the determinant of its positive semidefinite (PSD) kernel matrix $\det(K_S)$. Here,

$(K_S)_{ij}$ computes the kernel similarity between samples $s_i$ and $s_j$, typically derived from their embeddings. The diversity $\text{Div}(S)$ of $S$ is thus measured via this determinant.

Unlike prior works, we introduce a normalization factor to bound the diversity values within a specific range. The denominator is an approximate upper bound for the numerator, derived from the largest eigenvalue $\lambda_{\max}(K_\mathcal{U})$ of the kernel matrix over the entire ground set $\mathcal{U}$. The proposed normalized and submodular diversity term is:

$$\text{Div}(S) = \frac{\log \det(K_S + \epsilon I)}{\log(\lambda_{\max}(K_\mathcal{U}) + \epsilon)} \tag{3}$$

where the numerator is a log-determinant term stabilized by a small offset $\epsilon$. We set $\epsilon \geq 1$ to ensure $\text{Div}(S) \geq 0$. The function is submodular, meaning that adding an element to a smaller set yields a greater marginal gain in diversity, and a more diverse set generally corresponds to a larger $\text{Div}(S)$ score. Normalization prevents the unbounded $\log \det(\cdot)$ from dominating the objective $f$. Detailed proof of the submodularity and range of $\text{Div}(S)$ is provided in the Appendix A.1.3.

### 3.3. Adaptive Submodular Representativeness Term

The representativeness term aims to ensure that the selected coreset effectively represents the overall unlabeled pool $\mathcal{U}$. Intuitively, this means that each sample in $\mathcal{U}$ should be "covered" or well-approximated by at least one sample in the coreset, thereby preserving the key characteristics of the data distribution and yielding informative in-context examples. This concept aligns with the idea of selecting prototypes to summarize a dataset, as explored in prior works on data subset selection and active learning (Sener & Savarese, 2018; Wolf & Shashua, 2003). To operationalize this intuition, we assess the representativeness of a sample by synthesizing two complementary perspectives:

- **Density**: It should lie in a high-density region of the unlabeled data pool $\mathcal{U}$, ensuring it captures a substantial portion of the underlying data distribution.

- **Redundancy**: It should be geometrically distant from those already selected in $S$ to prevent information overlap and encourage non-neighbors within the coreset.

To formalize these concepts, we begin by defining a data-adaptive neighborhood scope (radius $r$), based on which the density score ($\rho_i$) measures how many other samples lie within distance $r$ (the larger the better), and the redundancy decay factor ($\gamma_i$) penalizes those who fall too close to already selected samples. These components are coupled together as the unified $\text{Rep}(S)$, which is monotone adaptive submodular. The proof is given in Appendix A.1.4.

$$\text{Rep}(S) = \sum_{s_i \in S} \rho_i \cdot \gamma_i \tag{4}$$

**Data-adaptive Neighborhood Radius $r$.** A robust radius $r$ is determined from the distribution of distances from each sample in $\mathcal{U}$ to its $k$-th nearest neighbor. To ensure each selected sample in the coreset covers a meaningful portion of the data, we scale $k$ inversely with the budget $m$: a smaller $m$ requires each selected sample to cover more data, hence $k$ should increase accordingly. Thus, we set $k = \lceil \sqrt{\kappa \cdot \frac{n}{m}} \rceil$, where $\kappa$ is a scaling factor that determines the value of the neighborhood size parameter $k$. Empirical studies in kNN-based density estimation suggest that setting $\kappa$ within the range $[10, 50]$ ensures stable local statistics with a moderate neighborhood size while avoiding excessive bias (Biau & Devroye, 2015).

Let $\mathbf{x}$ denote the embedding of sample $x \in \mathcal{U}$ and $\mathbf{x}^{(k)}$ for its $k$-th nearest neighbor. Motivated by robust statistics (Hampel, 1974), we estimate $r$ robustly as the median distance of each sample to its $k$-th nearest neighbor across all samples, as it is less sensitive to outliers and long-tailed distributions than the mean. Consequently, the definition of the neighborhood—and thus the local density estimate—adapts automatically to the inherent data distribution and the chosen scale parameter $k$, which can itself be linked to the available annotation budget.

$$r = \text{Med}\left(\{\|\mathbf{x} - \mathbf{x}^{(k)}\|_2 : x \in \mathcal{U}\}\right) \tag{5}$$

Based on $r$, the criteria $\rho_i$ measures the surrounding unlabeled samples for a sample $s_i$, where $\mathbf{1}\{\cdot\}$ is the indicator function, and $\gamma_i$ is computed by the current coreset.

$$\rho_i = \frac{\sum_{x_j \in \mathcal{U}} \mathbf{1}\{\|\mathbf{s}_i - \mathbf{x}_j\|_2 < r\}}{\max_{x_k \in \mathcal{U}} \sum_{x_j \in \mathcal{U}} \mathbf{1}\{\|\mathbf{x}_k - \mathbf{x}_j\|_2 < r\}} \tag{6}$$

$$\gamma_i = \exp\left(-\sum_{s_j \in S, j < i} \mathbf{1}\{\|\mathbf{s}_i - \mathbf{s}_j\|_2 < r\}\right) \tag{7}$$

Here $j < i$ indicates that only previously selected samples following the greedy order that lie within the radius $r$ of $s_i$ contribute to the decay. This formulation reflects an information-theoretic intuition: under a Poisson-like process for sample appearances in local neighborhoods, the marginal information provided by a new sample decays exponentially as its neighborhood becomes more populated.

### 3.4. Adaptive Estimation of Balance Factor $\alpha$

A critical component of our method is the balance factor $\alpha$, which controls the trade-off between diversity and representativeness. We propose an adaptive estimation of $\alpha$ based on two key considerations: magnitude scaling and budget awareness. Specifically, we define $\alpha$ as:

$$\alpha = \phi(m) \cdot \frac{\mathbb{E}_{t=1,\dots T}[\Delta \text{Div}_t]}{\mathbb{E}_{t=1,\dots T}[\Delta \text{Rep}_t]} \tag{8}$$

where $\phi(m)$ is a budget-aware term and the ratio of expectations acts as a rescaling term that balances the typical magnitudes of the incremental diversity $\Delta\mathrm{Div}_t$ and representativeness $\Delta\mathrm{Rep}_t$. To be noted here, after estimation, $\alpha$ is set as a fixed positive constant in coreset selection to ensure $f$ is adaptive submodular.

**Budget-adaptive $\phi(m)$.** Under a tight budget, maximizing representativeness is paramount for capturing the core data distribution. As the budget increases, diversity becomes more critical to avoid redundancy (as evidenced in Section 4.3). To automate this shift in priority, we design the budget-aware term $\phi(m)$ to depend on the current budget $m$ relative to the dataset size $n$. We introduce an empirical constant $C = 100$ to map the dataset size onto a range comparable to typical ICL budgets (Su et al., 2023; Zhang et al., 2024b), and apply clipping to prevent representativeness from being overly amplified when $n$ is very large.

$$\phi(m) = \exp\left(\mathrm{clip}\left(\frac{\lceil n/C\rceil - m}{C}, -1, 1\right)\right) \quad (9)$$

This formulation yields $\phi(m) \in (e^{-1}, e^1)$, ensuring a smooth, monotonic transition across budget range. Namely, $\phi(m) > 1$ for smaller budgets to boost representativeness, while $\phi(m) \leq 1$ for larger budgets to favor diversity.

**Magnitude Scaling.** To balance the typical magnitudes of the diversity gain $\Delta\mathrm{Div}_t$ and the representativeness gain $\Delta\mathrm{Rep}_t$, we estimate a scaling factor from their empirical averages. We obtain these averages by performing a preliminary greedy selection sequence.

Specifically, in each step $t$, let $\Delta\mathrm{Div}_t = \mathrm{Div}(S_{t-1}\cup\{s_t\}) - \mathrm{Div}(S_{t-1})$ define the marginal gain in diversity; $\Delta\mathrm{Rep}_t$ is defined analogously for representativeness. We perform $T$ steps of greedy selection based on the unweighted sum of marginal gains, $\Delta f_t = \Delta\mathrm{Div}_t + \Delta\mathrm{Rep}_t$. The expectations $\mathbb{E}_{t\in[1,T]}[\cdot]$ in Eq. (8) are the empirical averages over $T$ steps.

We set $T = 2m$, a larger value than the budget, as this longer preliminary sequence provides stable estimates of the average gains while remaining computationally inexpensive (details in Appendix A.3.6).

## 4. Experiments

### 4.1. Experimental Setup

**Using the Selected Coreset for ICL.** For each task, we first select a coreset $S$ from the unlabeled pool and obtain annotations for these samples. In practice, we use the available labels from the training set. The labeled coreset is then used to construct in-context learning (ICL) demonstrations. The ICL prompt for each test instance is populated with retrieved and ranked examples. This setup is compatible with arbitrary downstream example retrieval strategies (Min

et al., 2022; Ye et al., 2023; Liu et al., 2022).

**Datasets and tasks.** We evaluate our method on 7 typical NLP datasets: MRPC (Dolan et al., 2004), SST-5 (Socher et al., 2013), CommonsenseQA (CQA) (Talmor et al., 2019), ARC (Bhakthavatsalam et al., 2021), MNLI (Williams et al., 2018), SICK (Marelli et al., 2014), and e2eNLG (Novikova et al., 2017). It covers tasks including classification, multi-choice QA, natural language inference, and generation. We use the original train/test splits. The training sets are treated as the unlabeled pool $\mathcal{U}$ for coreset selection, while the test sets serve for the unseen evaluation. We use the development set for the evaluation of MNLI and CQA due to the unavailability of test labels. We report ROUGE-L (Lin, 2004) for the generation tasks (i.e., e2eNLG), and accuracy for the other tasks. Details are provided in Appendix A.2.

**Baselines.** We compare our method with the Random and SOTA coreset selection baselines, including IDEAL (Zhang et al., 2024b), RDSS (Shao et al., 2024), and InfoMax (unsupervised version) (Tan et al., 2025). We use their provided official codes. For a fair comparison, all baselines use the same backbone, prompt template, and budget; only the selected examples differ. This ensures that the performance differences are solely attributable to the coreset selection methods. The prompts used for each task are provided in Appendix A.2.

**Annotation Budget and Models.** Following prior work (Zhang et al., 2024b; Su et al., 2023), we consider two annotation budgets: 18 and 100. For the 18 setting, all selected examples are used as the ICL prompt to directly evaluate the quality of the coreset, while for 100, we retrieve Top-18 for each test sample (Liu et al., 2022). Demonstrations in the prompt are ranked in ascending order of their similarity. We use LLaMA3-8B-Instruct (Team, 2024) as the backbone, and additionally evaluate on LLaMA2-7B-Chat-hf [1] (Touvron et al., 2023). We use the Sentence-BERT (Reimers & Gurevych, 2019) model to obtain the text embeddings that serve as the feature representations for coreset selection.

### 4.2. Main Results

We present the main comparison results in Table 1. PaCS-ICL achieves the best results under most experimental settings. Our findings can be summarized in three key points.

*PaCS-ICL is budget-robust, and more advantageous under smaller budgets.* Our method delivers consistent improvements across different annotation budgets. The performance gain over the best baseline is larger when the budget is tight

---

[1] https://huggingface.co/meta-llama/Llama-2-7b-chat-hf

*Table 1.* Main results (Accuracy/Rouge-L) of coreset selection for ICL across 7 datasets on LLaMA3-8B and LLaMA2-7B. For each budget $m$, the best result is **bolded**.

| Backbone | Budget | Methods | MRPC | SST-5 | CQA | ARC | MNLI | SICK | e2eNLG | avg. |
|---|---|---|---|---|---|---|---|---|---|---|
| LLaMA3-8B | 18 | Random | 63.3 | 35.7 | 77.8 | 79.0 | 69.1 | 64.8 | 34.0 | 60.5 |
| | 18 | RDSS | 66.1 | 36.8 | 77.3 | 79.7 | 72.6 | 67.5 | 29.7 | 61.4 |
| | 18 | IDEAL | 66.9 | 36.6 | 77.8 | 79.7 | **73.9** | 58.3 | 34.8 | 61.1 |
| | 18 | InfoMax | 65.3 | 36.1 | 76.7 | 80.5 | 71.4 | 65.6 | 35.1 | 61.5 |
| | 18 | **PaCS-ICL** | **69.3** | **38.2** | **80.0** | **81.0** | 73.5 | **75.3** | **35.7** | **64.7** |
| | 100 | Random | 64.8 | 37.0 | 77.9 | 81.3 | 68.0 | 72.8 | 29.8 | 61.7 |
| | 100 | RDSS | 68.1 | 37.1 | 78.6 | 80.7 | 65.6 | 73.3 | 29.9 | 61.9 |
| | 100 | IDEAL | 68.1 | 36.4 | 77.8 | 80.4 | 67.9 | 72.0 | 31.3 | 62.0 |
| | 100 | InfoMax | 63.9 | 37.1 | 78.1 | 80.6 | 70.4 | **73.4** | **35.7** | 62.7 |
| | 100 | **PaCS-ICL** | **68.7** | **37.6** | **79.4** | **81.6** | **72.4** | **73.4** | 33.6 | **63.8** |
| LLaMA2-7B | 18 | Random | 68.7 | 46.0 | 56.9 | 56.2 | 49.0 | 61.1 | 37.4 | 52.5 |
| | 18 | RDSS | 68.6 | 46.4 | 59.0 | 56.5 | 43.0 | 62.6 | 31.5 | 52.5 |
| | 18 | IDEAL | 64.6 | 46.3 | 57.7 | 56.6 | 50.8 | 64.3 | 36.4 | 53.8 |
| | 18 | InfoMax | 69.2 | 41.0 | 58.1 | 56.5 | 53.5 | 62.7 | 34.9 | 53.7 |
| | 18 | **PaCS-ICL** | **71.0** | **49.5** | **60.2** | **56.7** | **57.5** | **67.1** | **42.1** | **57.7** |
| | 100 | Random | 67.5 | 47.0 | 58.3 | 56.1 | 54.0 | 60.0 | 35.0 | 54.0 |
| | 100 | RDSS | 70.1 | 47.3 | 57.6 | 57.5 | 51.6 | 63.7 | 35.1 | 54.7 |
| | 100 | IDEAL | 67.9 | 45.9 | 58.0 | 55.9 | 54.4 | 64.5 | 34.6 | 54.5 |
| | 100 | InfoMax | 67.3 | 43.3 | 59.5 | 54.9 | 55.9 | 64.4 | 34.6 | 54.3 |
| | 100 | **PaCS-ICL** | **71.5** | **49.2** | **60.1** | **58.0** | **56.6** | **65.3** | **38.4** | **57.0** |

(3.6 on average with $m = 18$) compared to a more relaxed setting (1.9 on average with $m = 100$). This indicates that PaCS-ICL is particularly effective when each selected example needs to be highly informative. This advantage stems from its balanced and non-redundant coverage, as discussed later (Section 4.5). It also benefits from the budget-aware trade-off between diversity and representativeness, as further elaborated in Section 4.3. We provide extended experimental results on budget robustness in Appendix A.3.9.

*Our method is dataset-robust.* Tasks requiring deeper semantic understanding are more sensitive to coreset selection, such as SICK, MRPC, and MNLI. The performances by different selection methods show high variances on these tasks, such as 6.8 for MNLI and 26.2 for SICK. This aligns with the observation that ICL performance is sensitive to demonstration quality on semantically complex tasks (Zhou et al., 2024). PaCS-ICL delivers the most stable performance across this spectrum.

*Our method is backbone-robust.* The performance on different backbones varies on tasks. The results on LLaMA3-8B significantly outperform on LLaMA2-7B on reasoning-intensive tasks, including QA and NLI, which illustrate LLaMA3's superior natural language understanding and reasoning abilities (Team, 2024). In contrast, LLaMA2 exhibits a slight advantage over LLaMA3 in classification (MRPC and SST-5) and text generation tasks (e2eNLG). PaCS-ICL achieves top performances on both backbones across tasks, demonstrating its robustness as a model-agnostic selector.

### 4.3. Ablation Study

We compare the full PaCS-ICL objective against its variants that remove the diversity term $\text{Div}(S)$, the representativeness term $\text{Rep}(S)$, and the balancing factor $\alpha$, respectively. We report the ablation results in Table 2.

*The budget-dependent balance between $\text{Div}(S)$ and $\text{Rep}(S)$ ensures stable performances.* Under a tight budget of 18, removing $\text{Rep}(S)$ degrades performance more than removing $\text{Div}(S)$. The trend reverses under a larger budget of 100, where diversity becomes more important. This confirms the premise: a smaller annotation budget requires higher representativeness to select the most informative samples, while a larger budget requires higher diversity to reduce local redundancy.

*The balance factor $\alpha$ corrects the scale imbalance between the two terms.* Without $\alpha$, the $\text{Rep}(S)$ dominates the selection. As shown in the table, the variant without $\alpha$ performs similarly to the variant without the $\text{Div}(S)$ term. Further empirical studies show that the raw marginal gains from $\text{Rep}(S)$ are often larger than those from $\text{Div}(S)$, such that $\text{Rep}(S)$ dominates the selection. The factor $\alpha$ rescales $\text{Rep}(S)$ to be comparable with $\text{Div}(S)$, which prevents one term from overwhelming the other.

### 4.4. Robustness to Retrieval Strategies of ICL

The selected coreset is ultimately utilized within a downstream retrieval mechanism to construct ICL prompts. A key

*Table 2.* Ablation study of PaCS-ICL components on LLaMA3-8B.

| Budget | Methods | MRPC | SST-5 | CQA | ARC | MNLI | SICK | e2eNLG |
|--------|---------|------|-------|-----|-----|------|------|--------|
| 18 | PaCS-ICL full | **69.3** | **38.2** | **80.0** | **81.0** | **73.5** | 75.8 | **35.7** |
| | $w/o \operatorname{Div}(S)$ | 69.0 | 34.1 | 78.6 | 80.5 | 70.8 | **77.7** | 32.3 |
| | $w/o \operatorname{Rep}(S)$ | 58.3 | 33.5 | 77.1 | 79.9 | 73.4 | 55.4 | 25.2 |
| | $w/o \alpha$ | 69.0 | 34.1 | 78.9 | 80.5 | 70.8 | 77.4 | 32.0 |
| 100 | PaCS-ICL full | **68.7** | **37.6** | **79.4** | **81.6** | 72.4 | **73.4** | **33.6** |
| | $w/o \operatorname{Div}(S)$ | 64.8 | 36.5 | 78.8 | 81.2 | 72.4 | **73.4** | 33.0 |
| | $w/o \operatorname{Rep}(S)$ | 68.5 | 37.5 | 79.1 | 80.6 | **73.6** | 61.8 | 29.5 |
| | $w/o \alpha$ | 65.1 | 35.8 | 78.8 | 81.3 | 72.5 | 71.7 | 32.5 |

practical consideration is whether the quality of the coreset is contingent upon a specific retrieval strategy. To evaluate this, we test coresets selected by the strongest baseline (InfoMax) and our PaCS-ICL under three distinct retrieval paradigms: Random (Min et al., 2022), Diversity-based (Ye et al., 2023), and Similarity-based (Liu et al., 2022). We use LLaMA3-8b with the budget $m = 100$. Results are summarized in Table 3.

*The coreset of PaCS-ICL shows higher intrinsic quality.* Under Random retrieval, which evaluates performance on randomly selected test samples, PaCS-ICL achieves higher performance than InfoMax. This indicates that the coreset maintains strong overall quality when there is no guidance from a specific retrieval method.

*The coreset of PaCS-ICL provides higher coverage on test samples.* When using relevance-focused retrieval (e.g., Similarity-based retrieval), PaCS-ICL surpasses InfoMax across datasets, suggesting that, for individual test samples, its coreset contains examples that better cover the input space and provide informative signals. Under Diversity-based retrieval, PaCS-ICL achieves higher performance than InfoMax, indicating that the coreset provides informative samples across queries with diverse characteristics.

In summary, PaCS-ICL shows robust performance across retrieval paradigms. Its core strength lies in achieving an intrinsic balance between diversity and representativeness through principled adaptive submodular co-optimization. It ensures that the selected subset contains both prototypical examples and a well-distributed set, reducing dependence on fine-tuning the downstream retrieval component.

### 4.5. Model Analysis

**Analysis on Process-Aware Coreset Selection.** To understand how our model works, we analyze the selection trajectories on the contribution of diversity and representativeness. Specifically, for each step $t = 1, \ldots, m$, we compute the proportion of the Rep term's marginal gain to the total gain, i.e., $\alpha \frac{\Delta \operatorname{Rep}_t}{\Delta f_t}$, where $\Delta f_t = \Delta \operatorname{Div}_t + \alpha \Delta \operatorname{Rep}_t$ This ratio quantifies the instantaneous balance between the

two terms. For comparison, we also show the trajectories for variants where the balancing factor $\alpha$ is omitted (red line), and where the budget-adaptive function $\phi(m)$ is removed (green line). As shown in Fig. 1, we present results from three datasets as examples; others are in Appendix A.3.2. It reveals several key findings:

*Rescaling in $\alpha$ balances representativeness and diversity.* Without the balancing factor $\alpha$ (red line), the representativeness term dominates the marginal gain throughout the selection process. This dominance arises from the mismatched values of the raw $\operatorname{Div}(S)$ and $\operatorname{Rep}(S)$ functions. It greatly biases the selection toward representativeness. The rescaling component of $\alpha$ (green line) brings the two terms to comparable magnitudes. As a result, their contribution ratio stabilizes around 0.5, indicating that diversity and representativeness exert similar influence on the selection.

*Budget-adaptive weighting $\phi(m)$ shifts the balance with budget.* Incorporating the budget-adaptive component $\phi(m)$ (blue line) further modulates the balance according to the annotation budget $m$. For smaller $m$, the trajectory shifts slightly above 0.5, placing more emphasis on representativeness to ensure adequate coverage under tight budgets. For larger $m$, it shifts slightly below 0.5, increasing the influence of diversity in reducing redundancy.

*Selection dynamically adapts to already selected samples.* The final balanced trajectory (blue line) exhibits two features. First, it shows a pronounced zigzag pattern. Since both $\operatorname{Rep}(S)$ and $\operatorname{Div}(S)$ are monotone submodular (or adaptive submodular), a single dominant term would produce a smooth, monotonic curve; the observed oscillation indicates active trade-offs between the two terms during selection. In addition, the $\operatorname{Rep}(S)$ contributes more in the early steps, while $\operatorname{Div}(S)$ becomes more influential later. This behavior arises naturally from the design of the representativeness term, which dynamically accounts for already selected samples and penalizes local redundancy. As more neighbors within radius $r$ are selected, the marginal gain of representativeness decays, allowing diversity to guide subsequent selections. This dynamic ensures the coreset achieves broad coverage with minimal redundancy.

*Table 3.* A comparison of the robustness of different retrieval strategies across datasets.

| Retrieval Method | UCS Method | MRPC | SST-5 | CQA | ARC | MNLI | SICK | e2eNLG | avg. |
|---|---|---|---|---|---|---|---|---|---|
| Random | InfoMax | 65.9 | 36.3 | 76.5 | 79.9 | **72.6** | 66.1 | 28.3 | 60.8 |
| | PaCS-ICL | **66.5** | **36.7** | **78.9** | **80.5** | 71.6 | **68.7** | **32.4** | **62.2** |
| Diversity | InfoMax | 64.4 | 35.3 | 77.6 | 80.6 | **72.6** | 64.7 | 32.3 | 61.1 |
| | PaCS-ICL | **67.2** | **37.6** | **78.8** | **81.1** | 70.3 | **65.7** | **33.9** | **62.1** |
| Similarity | InfoMax | 63.9 | 37.1 | 78.1 | 80.6 | 70.4 | **73.4** | **35.7** | 62.7 |
| | PaCS-ICL | **68.7** | **37.6** | **79.4** | **81.6** | **72.4** | **73.4** | 33.6 | **63.8** |

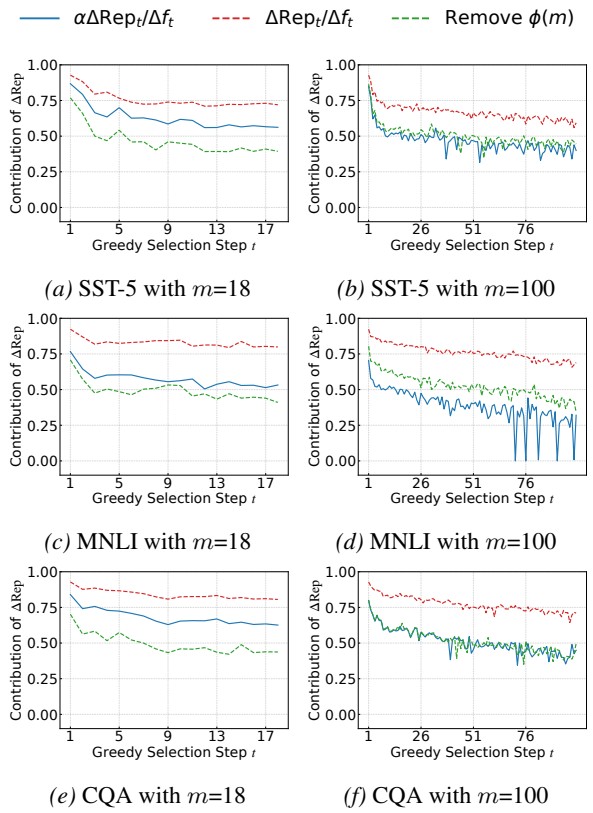

*Figure 1.* Contribution of the representativeness term during greedy selection in PaCS-ICL.

**Analysis of the Radius Adaptation** To validate our data-adaptive strategy for determining $r$ (Eq. 5), we conduct an ablation study comparing it with a fixed empirical radius and variants using different distribution quantiles. The results are summarized in Table 4. A fixed radius, e.g., $r = 0.3$, fails to generalize across datasets and embedding scales, resulting in unstable performance. Using distribution quantiles of the $k$-nearest neighbor distances is more adaptive; however, the choice of quantile significantly affects results. A lower quartile (0.25) yields an overly small radius, fragmenting the data manifold and reducing coverage, whereas an upper quartile (0.75) produces a radius that insufficiently penalizes local redundancy, encouraging clustered selections.

We adopt the median of the $k$-nearest neighbor distances

to define the global radius $r$, rather than the mean. This choice is motivated by robust statistics (Hampel, 1974), as the median is less sensitive to outliers and skewed distributions. In unsupervised settings, local distance distributions of embeddings can be skewed; using the median provides a stable and representative estimate of neighborhood scale. This ensures that the density term $\rho_i$ remains robust, allowing the representativeness term to effectively select samples from typical regions while the decay factor $\gamma_i$ suppresses redundant picks within neighborhoods of size $r$.

**Visualization of Coreset Selection.** To visualize the geometric structure of the selected samples, we perform t-SNE on the embeddings of the unlabeled samples. As shown in Fig. 2, the samples selected by different methods are highlighted in red. We use the MRPC dataset as a representative example. It can be observed that:

*PaCS-ICL effectively avoids neighbor redundancy.* Baseline methods, such as IDEAL (Zhang et al., 2024b) and Info-Max (Tan et al., 2025), tend to over-select samples from high-density regions, which leads to local redundancy. In contrast, PaCS-ICL incorporates a chosen sample-aware decay mechanism that explicitly suppresses such redundancy, thereby making more efficient use of the annotation budget.

*PaCS-ICL does not over-select edge samples.* Methods such as RDSS (Shao et al., 2024) and IDEAL often select outliers or edge samples, which have low representativeness. PaCS-ICL assigns lower priority to such samples, focusing instead on more representative regions of the data distribution.

In summary, PaCS-ICLachieves a more balanced coverage by selecting samples from both dense, representative regions and informative sparse areas, while avoiding over-representation of peripheral examples. This leads to a more efficient allocation of the annotation budget and improved quality of the selected coreset.

### 4.6. Efficiency Analysis

We analyze the computational complexity of PaCS-ICL and provide execution time comparisons with baseline coreset selection methods. PaCS-ICL consists of three main steps: (i) computing local densities for each candidate sample $x \in \mathcal{U}$, (ii) a probe greedy pass to calibrate $\alpha$, and (iii) the actual greedy coreset selection. Step (i) requires computing

*Table 4.* Performance comparison of different radius $r$ across tasks.

| Budget | Radius $r$ | MRPC | SST-5 | CQA | ARC | MNLI | SICK | e2eNLG | avg. |
|---|---|---|---|---|---|---|---|---|---|
| | $r = 0.3$ | 61.7 | 34.5 | 78.1 | 81.1 | 70.7 | 73.5 | 32.1 | 61.7 |
| 18 | $\text{quantile}_{0.25}$ | 63.2 | 35.9 | 79.2 | 80.0 | 68.8 | 62.8 | 32.4 | 60.3 |
| | $\text{quantile}_{0.75}$ | 66.4 | 35.2 | 78.8 | 80.5 | 70.3 | 63.7 | 35.3 | 61.5 |
| | Median | **69.3** | **38.2** | **80.0** | **81.0** | **73.5** | **75.3** | **35.7** | **64.7** |
| | $r = 0.3$ | 67.0 | 35.2 | 77.8 | 80.7 | 72.3 | 70.2 | 29.8 | 61.9 |
| 100 | $\text{quantile}_{0.25}$ | 67.7 | 36.9 | 78.5 | 81.0 | 71.1 | 67.7 | 31.2 | 62.0 |
| | $\text{quantile}_{0.75}$ | 62.9 | 34.7 | 78.5 | 78.5 | 70.9 | 68.6 | 33.2 | 61.0 |
| | Median | **68.7** | **37.6** | **79.4** | **81.6** | **72.4** | **73.4** | **33.6** | **63.8** |

• Unlabeled Samples    • Selected Samples

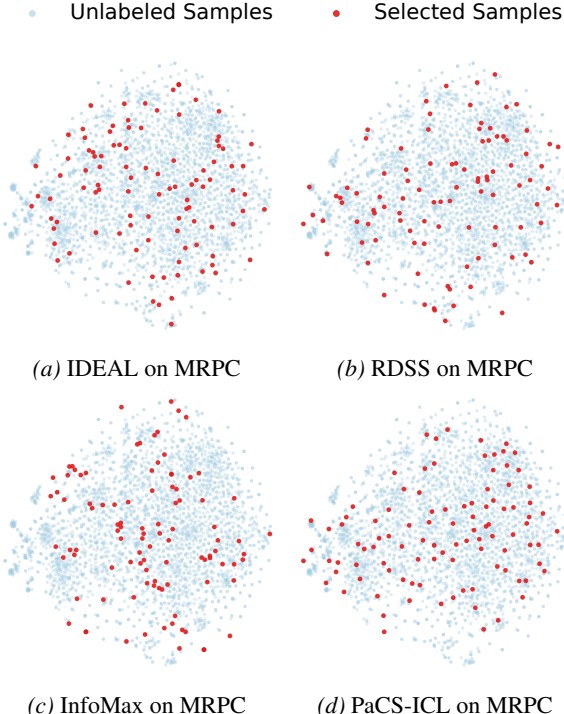

*(a) IDEAL on MRPC*    *(b) RDSS on MRPC*

*(c) InfoMax on MRPC*    *(d) PaCS-ICL on MRPC*

*Figure 2.* T-SNE visualization of the selected coreset by different methods. Red points indicate the selected samples.

*Table 5.* Execution time across different Budgets and Datasets.

| Budget | Method | SICK | CQA | e2eNLG |
|---|---|---|---|---|
| | | 5k | 10k | 30k |
| | IDEAL | >1h | >1h | >1h |
| 18 | RDSS | 26.8s | 234s | 3209s |
| | InfoMax | 1.8s | 7.2s | 79.4s |
| | **PaCS-ICL** | 14.0s | 32.0s | 134s |
| | IDEAL | >1h | >1h | >1h |
| 100 | RDSS | 28.2s | 193s | 2475s |
| | InfoMax | 1.7s | 7.6s | 78.8s |
| | **PaCS-ICL** | 191s | 432s | 1521s |

entire kNN graph for each candidate, resulting in significantly higher computational costs. Overall, the runtimes of RDSS, InfoMax, and PaCS-ICL remain within an acceptable range relative to the subsequent ICL inference stage. PaCS-ICL achieves substantial performance gains without introducing prohibitive computational overhead.

### 4.7. Scope Discussion

PaCS-ICL builds a query-independent global coreset in the select-then-annotation setting: a fixed set of samples is selected once from the unlabeled pool, without using any test queries or labeled data. This set can be annotated later and used for unseen downstream queries, in contrast to query-dependent methods that dynamically select per-query demonstrations.

Query-based methods excel when the query distribution is extremely long-tailed or highly heterogeneous, as they adapt the demonstration set to each individual query. However, this flexibility comes at the cost of requiring either LLM internals, e.g., gradients (Zhang et al., 2025c), or many labels to learn query–demonstration associations (Wang et al., 2025; Zhang et al., 2025b). In contrast, PaCS-ICL does not require labels or model internals, making it well-suited for cold-start scenarios. Its main limitation is that a fixed global coreset may be less precise for extremely long-tailed queries than a query-adaptive approach. Adapting to such cases may require pre-filtering candidates based on query neighborhoods or larger annotation budgets, which we consider promising future works. We provide empirical comparisons with query-based methods in Appendix A.3.7.

pairwise similarities over $\mathcal{U}$, which incurs $O(n^2)$ complexity in the worst case. However, it supports approximate search on large datasets (e.g., FAISS (Douze et al., 2026)) because Eqs. (5)–(7) only require radius- or top-$k$ neighbors. The empirical results show that GPU-parallelized full pairwise computation is faster than FAISS on tested datasets (Appendix A.3.3). Both steps (ii) and (iii) have a time complexity of $O(nm^2)$.

We report the execution time of the complete coreset selection process in Table 5, with experiments conducted on Tesla V100 32GB GPUs. As shown, InfoMax (Tan et al., 2025) is the fastest. PaCS-ICL is moderately efficient, being faster than RDSS (Shao et al., 2024) under tight budgets $m = 18$ and on larger datasets (e.g., 30k samples). IDEAL (Zhang et al., 2024b) requires multiple diffusion iterations over the

Our code is available[2].

## 5. Conclusion

In this work, we address the challenge of unsupervised coreset selection for few-shot in-context learning, where the goal is to select a small, informative subset from a large unlabeled pool under a tight annotation budget. We propose PaCS-ICL, a model-free framework that jointly optimizes diversity and representativeness through a monotone adaptive submodular objective. Our method incorporates a normalized DPP-based term to encourage diversity and a density-aware term with distance-based decay to capture representativeness while penalizing local redundancy. The balanced formulation enables efficient greedy selection with a theoretical approximation guarantee. Extensive experiments on 7 NLP benchmarks demonstrate that PaCS-ICL outperforms prior state-of-the-art coreset selection methods across different tasks, model scales, and annotation budgets. The ablation study validates the joint contribution of diversity and representativeness to the selection performance and demonstrates the importance of the balancing factor $\alpha$ in this process. Moreover, PaCS-ICL shows strong robustness under various retrieval strategies, highlighting its practical utility for real-world ICL applications. Our work provides a principled and efficient solution for unsupervised coreset selection, advancing the capability of LLMs to learn effectively from limited demonstrations. The future of this research lies in bridging query-independent coreset selection with query-based adaptation, particularly for extremely long-tailed or heterogeneous data. It also involves understanding the interplay between coreset selection objectives and LLMs' inductive biases, providing insights into when certain selection strategies improve performance across diverse tasks and model architectures.

## Acknowledgments

This work was supported by the National Natural Science Foundation of China under Grant No. 62376138.

## Impact Statement

This paper presents work whose goal is to advance the field of Machine Learning. There are many potential societal consequences of our work, none of which we feel must be specifically highlighted here.

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

# A. Appendix

## A.1. Theoretical Analysis

### A.1.1. STANDARD SUBMODULARITY

A set function $f : 2^{\mathcal{U}} \to \mathbb{R}$ is *submodular* if for any $A \subseteq B \subseteq \mathcal{U}$ and $x \in \mathcal{U} \setminus B$,

$$f(A \cup \{x\}) - f(A) \geq f(B \cup \{x\}) - f(B)$$

Intuitively, submodularity captures the *diminishing returns* property: adding an element to a smaller set provides at least as much gain as adding it to a larger set.

### A.1.2. ADAPTIVE SUBMODULARITY AND ADAPTIVE MONOTONICITY

In sequential decision making, the function value may depend on the order in which elements are selected. Following the framework of Golovin & Krause (2011), we consider *adaptive* set functions where the state $\psi$ represents the set of already selected elements (with their observed outcomes, which are deterministic in our setting). Let $\Delta(x \mid \psi) = f(\psi \cup \{x\}) - f(\psi)$ denote the marginal gain of adding element $x$ after having selected $\psi$.

- **Adaptive monotonicity:** $f$ is adaptively monotone if for every state $\psi$ and every element $x$, $\Delta(x \mid \psi) \geq 0$.

- **Adaptive submodularity:** $f$ is adaptively submodular if for any two states $\psi \subseteq \psi'$ (meaning $\psi'$ contains all information of $\psi$ and possibly more) and any element $x \notin \psi'$,

$$\Delta(x \mid \psi) \ \geq \ \Delta(x \mid \psi').$$

### A.1.3. RANGE AND SUBMODULARITY OF DIV(S)

Note that we define the Div(S) function as:

$$\mathrm{Div}(S) = \frac{\log \det(K_S + \epsilon I)}{\log(\lambda_{\max}(K_{\mathcal{U}}) + \epsilon)}$$

**Range of** $\mathrm{Div}(S)$**.** Let $\lambda_1(S), \ldots, \lambda_{|S|}(S)$ be the eigenvalues of $K_S + \epsilon I$. By basic linear algebra:

$$\det(K_S + \epsilon I) = \prod_{i=1}^{|S|} \lambda_i(S) \leq \prod_{i=1}^{|S|} \lambda_{\max}(K_{\mathcal{U}} + \epsilon I)$$
$$= (\lambda_{\max}(K_{\mathcal{U}}) + \epsilon)^{|S|}$$

$$\log \det(K_S + \epsilon I) \leq |S| \cdot \log(\lambda_{\max}(K_{\mathcal{U}}) + \epsilon)$$

Hence, $\mathrm{Div}(S) \leq |S|$. Since any positive semidefinite kernel, such as the RBF or inner product kernel, can be used to define $K_S$, we specifically employ the inner product kernel. Consequently, all eigenvalues of $K_S$ satisfy $\lambda_i \geq 0$.

$$\log \det(K_S + \epsilon I) = \sum_{i=1}^{|S|} \log(\lambda_i + \epsilon)$$
$$\geq \sum_{i=1}^{|S|} \log \epsilon = |S| \log \epsilon.$$

To ensure $\log \det(K_S + \epsilon I) \geq 0$, we choose $\epsilon \geq 1$. Combined with the upper bound above, we have

$$0 \leq \mathrm{Div}(S) \leq |S|$$

**Submodularity of** $\mathrm{Div}(S)$ It is known that $\log \det(K_S + \epsilon I)$ is submodular over positive definite matrices (Han et al., 2017). Since $\mathrm{Div}(S)$ is a positive linear transformation of $\log \det(K_S + \epsilon I)$, the marginal gain inequality still holds. Therefore, $\mathrm{Div}(S)$ is submodular.

A.1.4. RANGE AND ADAPTIVE SUBMODULARITY OF REP(S)

Recall that the representativeness function is defined as:

$$\text{Rep}(S) = \sum_{s_i \in S} \rho_i \cdot \gamma_i,$$

where the density scores $\rho_i \in [0, 1]$ and the redundancy-aware decay factors $\gamma_i = \exp\left(-\sum_{s_j \in S} \mathbf{1}\{\|\mathbf{s}_i - \mathbf{s}_j\|_2 < r\}\right) \in (0, 1]$.

**Range of** $\text{Rep}(S)$**.** For each selected sample $s_i \in S$, we have

$$0 < \rho_i \cdot \gamma_i \leq 1 \cdot 1 = 1.$$

Therefore, for a coreset of size $|S|$, we have

$$0 < \text{Rep}(S) = \sum_{s_i \in S} \rho_i \cdot \gamma_i \leq \sum_{i=1}^{|S|} 1 = |S|.$$

Hence, $\text{Rep}(S)$ lies in $[0, |S|]$, which is the same range as $\text{Div}(S)$.

**Adaptive Submodularity of** $\text{Rep}(S)$**.** Recall that $\text{Rep}(S) = \sum_{s_i \in S} \rho_i \cdot \gamma_i$, where $\gamma_i = \exp(-\sum_{s_j \in S, j < i} \mathbf{1}\{\|\mathbf{s}_i - \mathbf{s}_j\|_2 < r\})$ depends on the order of selection. In adaptive optimization, the state $\psi$ is the set of already selected elements (unordered), and the marginal gain of adding a new element $x$ given state $\psi$ is

$$\Delta_{\text{Rep}}(x \mid \psi) = \rho_x \cdot \gamma_x(\psi), \qquad \gamma_x(\psi) = \exp(-N_\psi(x)), \quad N_\psi(x) = \left|\{z \in \psi : \|\mathbf{z} - \mathbf{x}\|_2 < r\}\right|.$$

Although the total $\text{Rep}(S)$ depends on order, the marginal gain depends only on the unordered set $\psi$ because $N_\psi(x)$ counts neighbors regardless of their selection order.

*Adaptive monotonicity:* Since $\rho_x \geq 0$ and $\gamma_x(\psi) > 0$, we have $\Delta_{\text{Rep}}(x \mid \psi) \geq 0$ for any $\psi$ and any $x$.

*Adaptive submodularity:* For any two states $\psi \subseteq \psi'$ (i.e., $\psi'$ contains all elements of $\psi$ and possibly more) and any $x \notin \psi'$, we have $N_\psi(x) \leq N_{\psi'}(x)$ because additional elements can only increase the neighbor count, thus

$$\gamma_x(\psi) = \exp(-N_\psi(x)) \geq \exp(-N_{\psi'}(x)) = \gamma_x(\psi').$$

Multiplying by $\rho_x$ yields

$$\Delta_{\text{Rep}}(x \mid \psi) \geq \Delta_{\text{Rep}}(x \mid \psi').$$

This is the diminishing returns property required for adaptive submodularity. Therefore, $\text{Rep}(S)$ is adaptively monotone and submodular (Golovin & Krause, 2011).

A.1.5. GREEDY OPTIMIZATION AND APPROXIMATION GUARANTEE

In our framework, the overall objective $f(S) = \text{Div}(S) + \alpha \text{Rep}(S)$ is a non-negative linear combination of $\text{Div}(S)$ (standard monotone submodular) and $\text{Rep}(S)$ (adaptively monotone and adaptively submodular). Since adaptive submodularity and adaptive monotonicity are preserved under non-negative linear combinations (Golovin & Krause, 2011), $f$ is also adaptively monotone and adaptively submodular.

Therefore, the greedy selection in Eq. 2 satisfies the conditions of the adaptive greedy policy, which achieves a $(1 - 1/e)$ approximation guarantee:

$$\mathbb{E}\big[f(\psi_{\text{greedy}})\big] \geq \left(1 - \frac{1}{e}\right) \cdot \mathbb{E}\big[f(\psi^*)\big],$$

where $\psi_{\text{greedy}}$ is the final state after $m$ greedy selections and $\psi^*$ is the optimal state. In our deterministic setting, the expectation can be dropped. This guarantee justifies the greedy algorithm used in PaCS-ICL.

## A.2. Datasets and Prompts

Table 6 provides detailed statistics of the datasets used. These datasets were selected to represent a broad spectrum of challenges in In-Context Learning (ICL), ensuring that our evaluation is both diverse and comprehensive. The training set sizes for MNLI correspond to full dataset sampling, but for efficiency, we randomly sampled 5,000 samples to form the unlabeled pool $\mathcal{U}$ for coreset selection.

*Table 6.* Detailed statistics of datasets used in our experiments.

| Dataset | Task | Unlabeled size $n$ | Test size |
|---|---|---|---|
| MRPC (Dolan et al., 2004) | Text Similarity Classification | 3,668 | 1,725 |
| SST-5 (Socher et al., 2013) | Sentiment Classification | 8,544 | 2,210 |
| CQA (Talmor et al., 2019) | Multi-choice QA / Commonsense Reasoning | 9,741 | 1,221 |
| ARC (Bhakthavatsalam et al., 2021) | Multi-choice QA / Science Reasoning | 1,119 | 1,172 |
| MNLI (Williams et al., 2018) | Natural Language Inference | 5,000[*] | 9,815 |
| SICK (Marelli et al., 2014) | Natural Language Inference | 4,439 | 4,906 |
| e2eNLG (Novikova et al., 2017) | Text Generation | 33,525 | 4,693 |

[*] The training size for MNLI is based on full dataset sampling.

**Prompts.** The following prompts were used to evaluate our method for each task. For each test sample, we retrieve the 18 most similar demonstrations and arrange them in ascending order of similarity (Liu et al., 2022).

**MRPC:**

```
Determine whether the given two sentences have the same meaning.
Answer only with 'EQUIVALENT' or 'NOT EQUIVALENT'.

%% A demonstration is shown below
SENTENCE1:  "Yucaipa owned Dominick's before selling the chain to Safeway
in 1998 for $2.5 billion."
SENTENCE2:  "Yucaipa bought Dominick's in 1995 for $693 million and sold
it to Safeway for $1.8 billion in 1998."
ANSWER: NOT EQUIVALENT
...

%% test sample
SENTENCE1:  {text1}
SENTENCE2:  {text2}
ANSWER:
```

**SST-5:**

```
You are a sentiment classifier.  Only output one of these labels:  very
negative, negative, neutral, positive, very positive.  Do not output any
explanations or text.

%% A demonstration is shown below
SENTENCE: "The movie certainly has its share of clever moments and biting
dialogue, but there's just not much lurking below its abstract surface."
LABEL: neutral
...

%% test sample
```

```
SENTENCE: {text}
LABEL:
```

**ARC:**

```
You are solving a multiple-choice question.  Only output the letter of the
correct option.  Do not output any explanations.

%% A demonstration is shown below
QUESTION: "How does the tilt of Earth's axis and its rotation affect the
weather?"
OPTIONS:
A: "The tilt of Earth allows Earth to absorb all of the Sun's radiation as
it rotates."
B: "The tilt allows certain latitudes of Earth to be heated at a greater
rate while Earth rotates."
C: "The tilt of Earth allows Earth to rotate fast enough to allow surface
cooling to occur at night."
D: "The tilt allows energy to be evenly distributed throughout the
atmosphere while Earth rotates."
ANSWER: B
...

%% test sample
QUESTION: {question}
OPTIONS: {options}
ANSWER:
```

**CQA:**

```
You are solving a commonsense reasoning multiple-choice question.  Only
output the letter of the correct option.  Do not output any explanations.

%% A demonstration is shown below
QUESTION: " Where is the milky way?"
OPTIONS:
A: fog
B: candy bar shelf
C: stars
D: space
E: refrigerator
ANSWER: D
...

%% test sample
QUESTION: {question}
OPTIONS: {options}
ANSWER:
```

**MNLI:**

```
Determine the relationship between a Premise and a Hypothesis.  Only
output one of three labels:  ENTAILMENT, CONTRADICTION, NEUTRAL. Do not
```

```
output any explanations or text.

%% A demonstration is shown below
PREMISE: "The new rights are nice enough."
HYPOTHESIS: "Everyone really likes the newest benefits."
LABEL: ENTAILMENT
...

%% test sample
PREMISE: {premise}
HYPOTHESIS: {hypothesis}
LABEL:
```

**SICK:**

```
Determine the relationship between a Premise and a Hypothesis.  Only
output one of three labels:  ENTAILMENT, CONTRADICTION, NEUTRAL. Do not
output any explanations or text.

%% A demonstration is shown below
PREMISE: "A boy is wearing a white hat and is walking on the beach."
HYPOTHESIS: "A little kid is wearing a hat and is walking in the wet sand"
LABEL: NEUTRAL
...

%% test sample
PREMISE: {premise}
HYPOTHESIS: {hypothesis}
LABEL:
```

**e2eNLG:**

```
Your task is to generate a natural, fluent sentence that describes the
restaurant based on the given information.  Include all the information
provided.  Do not invent any details.

%% A demonstration is shown below
CONCEPTS: "name[Fitzbillies], eatType[coffee shop], food[Japanese],
priceRange[moderate], customer rating[1 out of 5], area[city centre]"
SENTENCE: "Fitzbillies is a Japanese coffee shop in the city center
for adults.  With a customer rating of 1 out of 5 the price range is
moderate."
...

%% test sample
CONCEPTS: {concepts}
SENTENCE:
```

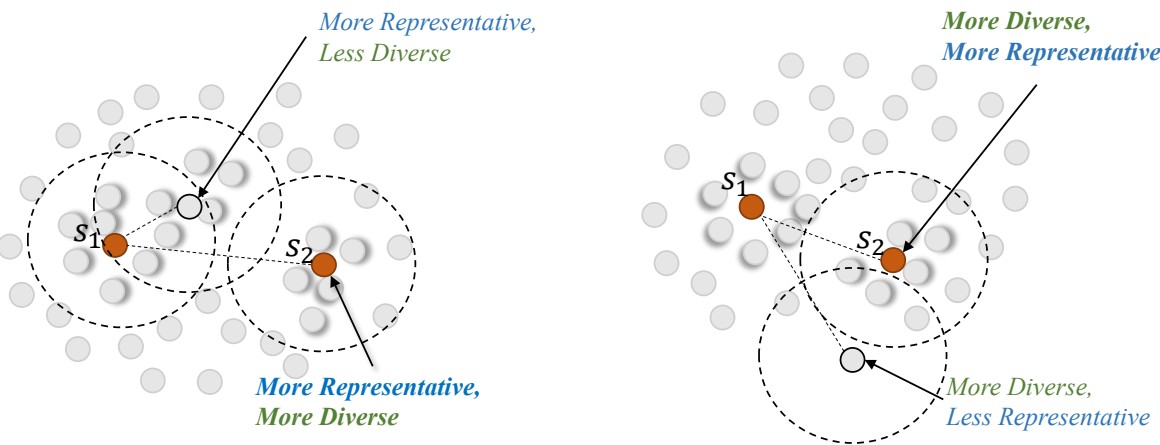

*Figure 3.* Schematic illustration of the synergistic selection of diversity and representativeness in PaCS-ICL.

## A.3. Supplementary Experiments

### A.3.1. SYNERGISTIC SELECTION OF DIVERSITY AND REPRESENTATIVENESS

Fig. 3 provides an intuitive visualization of the core selection mechanism. For clarity, it depicts a scenario after selecting a single initial sample $s_1$ (in orange) and illustrates the conceptual trade-off in choosing the next sample $s_2$. It shows that our method synergizes diversity and representativeness to mitigate local redundancy and peripheral samples. In practice, the selection of any new candidate is based on its marginal gain relative to the entire current set $S_{t-1}$ of previously chosen samples, not just the last one. This process-aware evaluation allows PaCS-ICL to balance broad coverage against local redundancy throughout the greedy selection process.

### A.3.2. SUPPLEMENTARY ANALYSIS OF BALANCING DIVERSITY AND REPRESENTATIVENESS.

This section provides a supplementary analysis of the interplay between diversity and representativeness during the greedy selection process for the remaining datasets: MRPC, ARC, SICK, and e2eNLG. Fig. 4 visualizes the marginal gain trajectories, extending our findings from Section 4.3. By analyzing these trajectories, we observe how the size of the unlabeled pool ($n$) influences the trade-off governed by the balancing factor $\alpha$. Across all datasets, the full PaCS-ICL objective (blue line) exhibits a non-monotonic, oscillating trajectory. This pattern indicates that the marginal gains of representativeness and diversity adjust in response to the selection context, in contrast to the static dominance of $\Delta\mathrm{Rep}$ when $\alpha$ is omitted (red line). This comparison highlights the role of our co-optimization framework.

A phenomenon associated with a large unlabeled pool size $n$ is observed. The e2eNLG dataset illustrates this: as shown in Fig. 4 (g) and (h), the contribution of $\Delta\mathrm{Rep}$ remains elevated throughout the selection process, even under the more permissive budget $m = 100$. This admits a statistical interpretation: when $n$ is large, each selected example should represent a broader region of the data manifold to achieve distributional coverage. Thus, the need for representativity appears to remain strong regardless of the absolute budget $m$. This is supported by evidence from Table 2: for e2eNLG at $m = 100$, removing $\mathrm{Rep}$ causes a performance drop from 33.6 to 29.5, suggesting that representative coverage is necessary for large-scale tasks.

### A.3.3. RUNTIME COMPARISON: FULL PAIRWISE VS. FAISS

We conducted additional experiments to compare the runtime of the density computation step with and without FAISS. The FAISS setup used `IndexIVFFlat + range_search`, with `nlist` = $\sqrt{n}$ and `nprobe` = $\sqrt{\texttt{nlist}}$. `No_Faiss` denotes computing the full pairwise similarity on $\mathcal{U}$. The results are shown in Table 7.

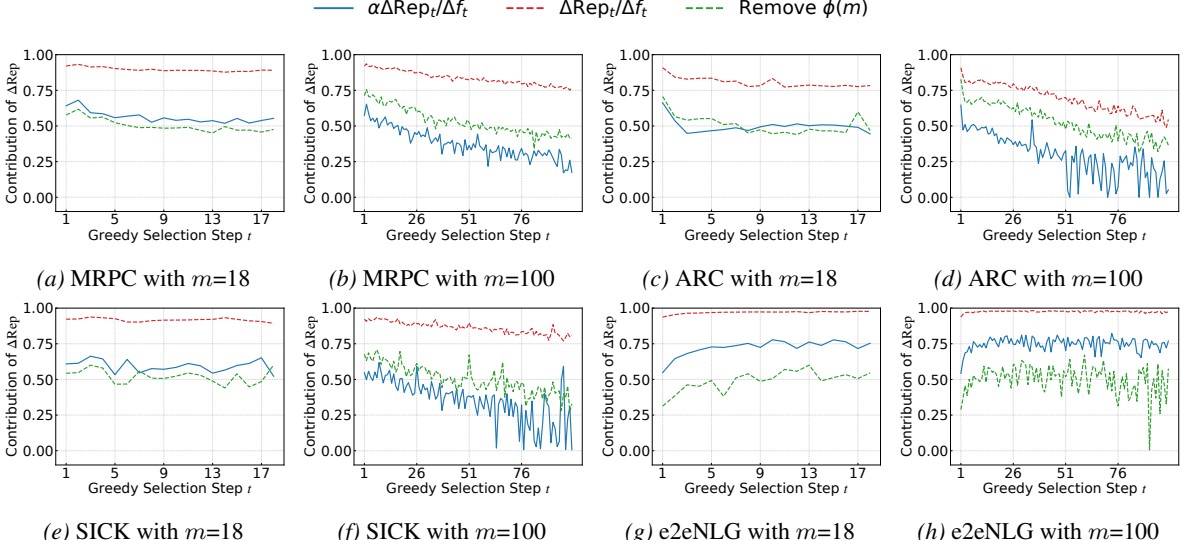

*Figure 4.* Trajectories of $\Delta \mathrm{Rep}(S)_t / \Delta f_t$ in the greedy selection process.

*Table 7.* Runtime (seconds) of density computation step on different datasets.

| Method | Device | SICK (5k) | CQA (10k) | e2eNLG (30k) | MNLI (392k) |
|---|---|---|---|---|---|
| No_Faiss | GPU | 0.6 | 0.6 | 3.4 | 214 |
| | CPU | 0.3 | 1.3 | 15.0 | 1920 |
| Faiss | CPU-only | 0.6 | 1.8 | 10.3 | 397 |

On CPU, FAISS is faster than `No_Faiss` for datasets of size $\geq$ 30k (e2eNLG and MNLI). `No_Faiss` with GPU parallelization is faster for all tested datasets. Hence we adopt GPU-based full pairwise computation for our experiments.

### A.3.4. SCALING WITH THE SIZE OF THE UNLABELED POOL

To investigate how the performance of PaCS-ICL scales with the number of available unlabeled samples, we conducted experiments on subsets of varying sizes from two datasets: CQA (up to 10k) and MNLI (up to 392k). For each dataset, we sampled subsets with increasing numbers of unlabeled examples and evaluated the quality of the selected coreset under two annotation budgets ($m = 18$ and $m = 100$). All results are averaged over three random seeds; the reported numbers are mean $\pm$ standard deviation. The results are summarized in Tables 8 and 9.

*Table 8.* Performance on CQA under different pool sizes and budgets.

| Budget | Method | 1k | 2k | 4k | 8k | 10k |
|---|---|---|---|---|---|---|
| 18 | Infomax | $77.9 \pm 0.65$ | $78.3 \pm 0.61$ | $78.1 \pm 1.21$ | $78.0 \pm 0.23$ | 76.7 |
| | PaCS-ICL | $80.3 \pm 0.92$ | $80.9 \pm 0.47$ | $81.9 \pm 0.72$ | $80.8 \pm 0.49$ | 80.0 |
| 100 | Infomax | $77.7 \pm 0.32$ | $77.5 \pm 0.28$ | $77.9 \pm 0.15$ | $77.5 \pm 0.41$ | 78.1 |
| | PaCS-ICL | $77.6 \pm 0.05$ | $77.9 \pm 0.11$ | $78.1 \pm 0.00$ | $78.8 \pm 0.85$ | 79.4 |

*Table 9.* Performance on MNLI under different pool sizes and budgets.

| Budget | Method | 1k | 2k | 4k | 10k | 20k | 40k | 392k |
|---|---|---|---|---|---|---|---|---|
| 18 | Infomax | $73.0 \pm 2.13$ | $71.6 \pm 2.40$ | $70.7 \pm 1.27$ | $72.6 \pm 0.10$ | $71.1 \pm 0.82$ | $70.2 \pm 2.08$ | 70.0 |
| | PaCS-ICL | $72.6 \pm 0.48$ | $70.6 \pm 2.30$ | $72.2 \pm 2.27$ | $72.9 \pm 1.90$ | $73.4 \pm 0.94$ | $70.9 \pm 0.20$ | 70.9 |
| 100 | Infomax | $71.1 \pm 0.50$ | $71.3 \pm 1.73$ | $71.5 \pm 0.89$ | $72.1 \pm 1.01$ | $71.8 \pm 0.32$ | $71.0 \pm 1.68$ | 71.4 |
| | PaCS-ICL | $72.6 \pm 0.61$ | $72.7 \pm 0.64$ | $72.0 \pm 0.79$ | $72.2 \pm 0.53$ | $72.4 \pm 0.66$ | $72.5 \pm 0.36$ | 73.8 |

For a low budget ($m = 18$), the performance of both methods tends to peak at a moderate pool size (e.g., 4k for PaCS-ICL

on CQA, 2k for Infomax on CQA, and 20k for PaCS-ICL on MNLI). Further increasing the unlabeled pool brings no additional gain, likely due to the fixed budget being too small to cover the increased data diversity. In contrast, for a higher budget ($m = 100$), enlarging the pool consistently improves performance: the gain is monotonic on CQA and shows an increasing trend on MNLI. These results suggest that, for larger unlabeled pools, using a larger annotation budget may be beneficial.

### A.3.5. LLAMA3 AS THE EMBEDDING MODELS

We additionally evaluate LLaMA3 as the text embedding model. Specifically, text representations are mean-pooled from the last-layer hidden states of each token. The budget is set to 18. The ICL performance of both InfoMax and our method is reported in Table 10. The results show that performance with Sentence-BERT is generally a bit higher than with LLaMA3 hidden states, but the differences are modest. We adopt Sentence-BERT as our embedding model because it is widely used in NLP tasks and its effectiveness has been validated in applications such as coreset selection, information retrieval, and text clustering. Moreover, its embedding dimension (768) is much smaller than that of LLaMA3 (4096), leading to higher computational efficiency.

*Table 10.* The ICL performance on different embedding models (Sentence-BERT and LLaMA3).

| Embedding Model | Method | MRPC | SST-5 | CQA | ARC | MNLI | SICK | e2eNLG |
|---|---|---|---|---|---|---|---|---|
| LLaMA3-hidden | InfoMax | 68.6 | 29.9 | 77.8 | 80.0 | 71.0 | 64.5 | 33.0 |
| | PaCS-ICL | 69.4 | 34.7 | 78.0 | 79.9 | 74.8 | 68.9 | 32.6 |
| Sentence-BERT | InfoMax | 65.3 | 36.1 | 76.7 | 80.5 | 71.4 | 65.6 | 35.1 |
| | PaCS-ICL | 69.3 | 38.2 | 80.0 | 81.0 | 73.5 | 75.3 | 35.7 |

### A.3.6. ABLATION ANALYSIS OF THE PROBE PROCESS

To assess the stability of the **probe** process, we inject Gaussian noise $\mathcal{N}(0, \sigma^2)$ with $\sigma = 0.1$ into the score of each candidate sample at each greedy step to influence the selection trajectory. This $\sigma$ corresponds to 53% of the *std.* observed in the noise-free probe process. Over 10 independent runs, we report the *mean* and *variance* of the estimated magnitude ratio $\frac{\mathbb{E}_{t=1:T}\left[\Delta \mathrm{Div}_t\right]}{\mathbb{E}_{t=1:T}\left[\Delta \mathrm{Rep}_t\right]}$ across runs. We vary the probe length $T$ from $0.5m$ to $3m$ and present the results in Table 11.

*Table 11.* Analysis of the **Probe** process under Gaussian noise. Reported as mean / variance of the estimated magnitude ratio over 10 runs.

| Dataset | $0.5m$ | $1m$ | $1.5m$ | $2m$ | $2.5m$ | $3m$ |
|---|---|---|---|---|---|---|
| MRPC | 0.11/1.0e−4 | 0.13/3.0e−4 | 0.15/2.0e−4 | 0.16/2.0e−4 | 0.18/2.0e−4 | 0.20/2.0e−4 |
| SST-5 | 0.23/3.0e−3 | 0.31/2.0e−3 | 0.38/2.0e−3 | 0.42/3.0e−3 | 0.47/3.0e−3 | 0.51/3.0e−3 |
| CQA | 0.17/2.4e−3 | 0.21/1.0e−3 | 0.27/7.4e−3 | 0.31/1.3e−3 | 0.33/1.7e−3 | 0.35/1.4e−3 |
| SICK | 0.09/3.0e−4 | 0.10/3.0e−4 | 0.12/6.0e−4 | 0.13/2.0e−4 | 0.14/5.0e−4 | 0.15/4.0e−4 |

*Table 12.* Ablation on the probe length $T$.

| Dataset | $0.5m$ | $1m$ | $1.5m$ | $2m$ | $2.5m$ | $3m$ |
|---|---|---|---|---|---|---|
| MRPC | 68.2 | 68.3 | 69.1 | 69.2 | 69.2 | 68.1 |
| SST-5 | 38.2 | 38.2 | 38.2 | 38.2 | 37.0 | 37.0 |
| CQA | 80.0 | 80.0 | 80.0 | 80.0 | 80.0 | 80.0 |
| ARC | 81.0 | 81.0 | 81.0 | 81.0 | 81.0 | 81.0 |
| SICK | 75.3 | 75.3 | 75.3 | 75.3 | 75.3 | 74.2 |
| MNLI | 73.2 | 73.2 | 73.5 | 73.5 | 73.5 | 73.5 |
| e2eNLG | 35.6 | 35.9 | 35.4 | 35.7 | 35.9 | 35.3 |

As shown in Table 11, the estimated magnitude ratio $\frac{\mathbb{E}_{t=1:T}\left[\Delta \mathrm{Div}_t\right]}{\mathbb{E}_{t=1:T}\left[\Delta \mathrm{Rep}_t\right]}$ increases with $T$, indicating that the contribution of representativeness diminishes as the probe extends. This occurs because $\Delta \mathrm{Rep}$ decays exponentially as different high-density regions become saturated in coverage, while $\Delta \mathrm{Div}$ remains stable. The selection process radiates from high-density regions

outward beyond the radius $r$. Moreover, the very small variance across runs demonstrates that the probe estimate is highly stable against noise.

We further conducted an ablation study on the ICL performance with different probe lengths $T$, as shown in Table 12. The results show that our method is insensitive to the choice of $T$ within a broad range, with the optimal $T$ typically around $2m$. Our motivation is to set $T$ such that $m < T \ll n$ focusing on "active samples" — those with a high potential of being selected in the final selection process. Based on the ablation results, we choose $T = 2m$ for all main experiments.

### A.3.7. EMPIRICAL COMPARISON WITH A QUERY-BASED METHOD: GRADERE

We compare PaCS-ICL against GradeRE (Zhang et al., 2025c), a recent query-based method that uses LLM gradients to estimate the influence of candidate demonstrations on a query set. We evaluate on CQA, ARC, MRPC, and SST-5. For GradeRE, we randomly sample 500 labeled examples from the original training set as the query training set, and the remaining labeled examples serve as candidates. The number of candidate subsets is 500, and the number of anchor sets is 1. We test two demonstration counts: 5 and 18. The sizes of the candidate subset and anchor set match the demonstration count.

The results are shown in Fig. 5. Across all settings, PaCS-ICL achieves better results, despite using a far smaller labeling budget compared to the supervised setting. These results suggest that for tasks with few and balanced classes, a small global coreset can be more effective than a query-dependent method that demands many labels.

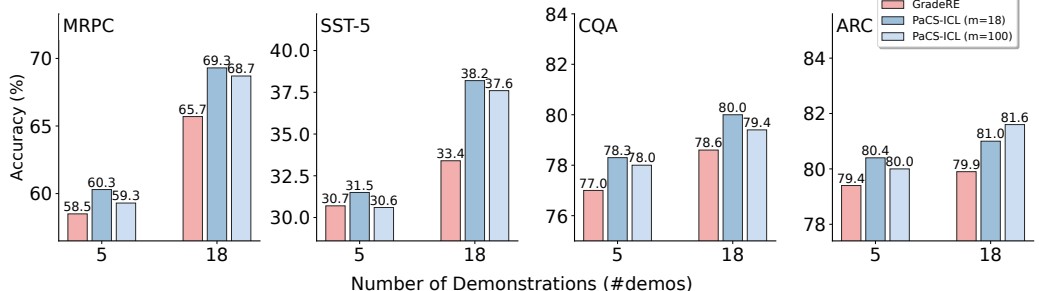

*Figure 5.* Comparison with GradeRE across tasks.

### A.3.8. VISUALIZATION OF THE GEOMETRY STRUCTURE OF THE SELECTED SAMPLES.

We provide additional t-SNE visualizations illustrating the geometric structure of the coresets selected by different methods on other datasets. Fig. 6 presents results on the ARC dataset for budgets 18 and 100, comparing IDEAL, RDSS, InfoMax, and our PaCS-ICL. These visualizations corroborate the observations detailed in Section 4.5 of the main text. The samples selected by PaCS-ICL exhibit a more balanced and widespread distribution within the embedding space compared to those chosen by baseline methods.

### A.3.9. BUDGET ROBUSTNESS ANALYSIS

*Table 13.* Performance comparison between InfoMax and PaCS-ICL under different **Budgets** on MRPC and SST-5 datasets.

| **Dataset** | **MRPC** | | | | | | | **SST-5** | | | | | | |
|---|---|---|---|---|---|---|---|---|---|---|---|---|---|---|
| **Budget** | 30 | 40 | 50 | 60 | 70 | 80 | 90 | 30 | 40 | 50 | 60 | 70 | 80 | 90 |
| InfoMax | 63.7 | 65.4 | **67.5** | 66.0 | 65.6 | 65.3 | 66.4 | 37.6 | 36.3 | 35.3 | 35.7 | 36.1 | 37.4 | 35.6 |
| PaCS-ICL | **67.3** | **67.8** | 66.6 | **66.5** | **65.7** | **65.8** | **67.7** | **38.3** | **37.7** | **36.5** | **38.8** | **38.1** | **38.0** | **37.7** |

We compare PaCS-ICL against the strongest baseline, InfoMax (Tan et al., 2025), across a wider range of budgets $m \in \{30, 40, 50, 60, 70, 80, 90\}$. Experiments are conducted using LLaMA3-8B on the MRPC and SST-5 datasets. The results are given in Table 13. The results confirm that PaCS-ICL is not optimized for a single operating point but provides an effective algorithm for coreset selection across a practical range of annotation budgets.

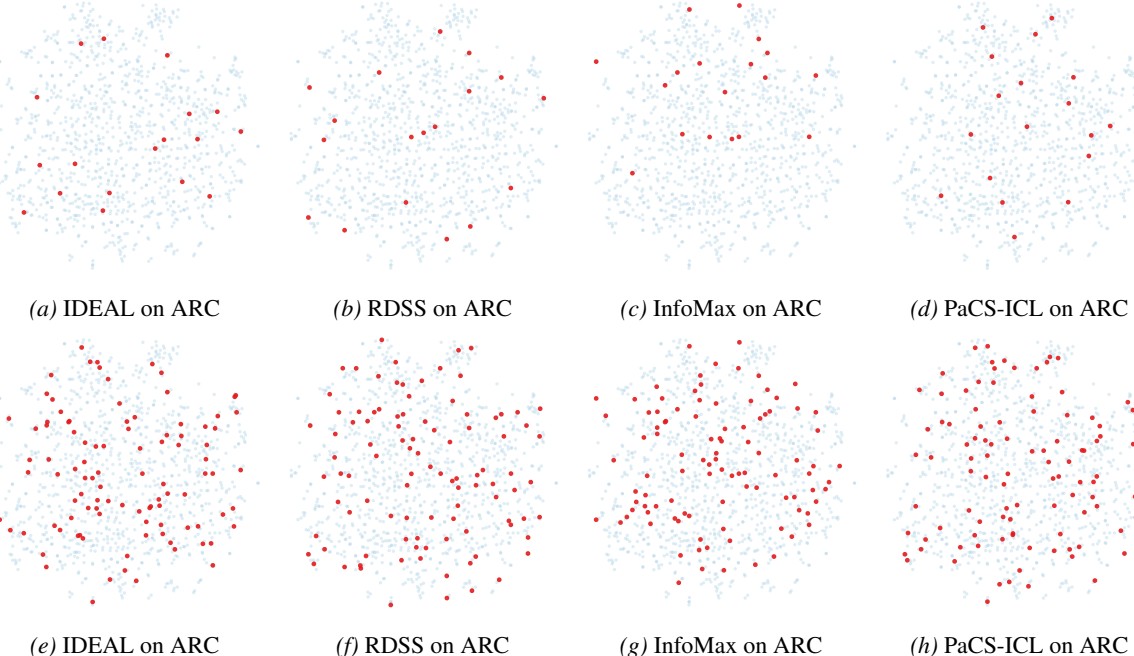

| | | | |
|---|---|---|---|
| *(a)* IDEAL on ARC | *(b)* RDSS on ARC | *(c)* InfoMax on ARC | *(d)* PaCS-ICL on ARC |
| *(e)* IDEAL on ARC | *(f)* RDSS on ARC | *(g)* InfoMax on ARC | *(h)* PaCS-ICL on ARC |

*Figure 6.* T-SNE visualization of the selected coreset by different methods. Red points indicate the selected samples.

### A.4. More Discussion on Future Work

*About the Geometric Faithfulness of Sample Embeddings.* Most of the current coreset selection approaches rely on pre-trained embeddings to define a geometric representation of the data. This introduces two related assumptions: that the embedding space captures semantic similarity faithfully, and that coverage in this space corresponds to effective in-context examples. While these assumptions generally hold for high-quality embeddings (e.g., Sentence-BERT), deviations in embedding geometry or biases in the embedding model may propagate into the selected coreset. Future work could explore augmenting geometric selection with lightweight, task-aware signals or jointly adapting embeddings to better align with coreset selection objectives.

*On the Coreset Selection Problem from an Incremental Unlabeled Pool.* The current formulation treats the unlabeled pool $\mathcal{U}$ as static. In real-world interactive settings, new data may arrive continuously. Extending PaCS-ICL to a streaming or incremental setting, where the coreset can be updated efficiently without recomputing from scratch, would broaden its applicability.

