# OpenReview forum: "Unsupervised Process-Aware Coreset Selection for In-Context Learning"
_ICML.cc/2026/Conference — ICML 2026 regular_

### Official Review · Reviewer_nLf8 · 2026-03-07

**Soundness:** 2
**Presentation:** 3
**Significance:** 3
**Originality:** 3
**Overall Recommendation:** 4
**Confidence:** 4

**Summary:**

This paper addresses the challenge of Unsupervised Coreset Selection (UCS) for few-shot In-Context Learning (ICL). The objective is to select a highly informative, small subset of examples from a large unlabeled pool to serve as effective few-shot demonstrations for Large Language Models (LLMs). Recognizing that existing model-free methods struggle to balance data representativeness against sample diversity (often resulting in redundant selections from high-density regions), the authors propose PaCS-ICL (Process-Aware Coreset Selection for ICL). PaCS-ICL frames coreset selection as the maximization of a unified, monotone submodular objective that explicitly co-optimizes diversity and representativeness using a greedy algorithm. The core contribution is a dynamic, process-aware scoring mechanism that prioritizes high-density typical samples while penalizing redundant candidates near already-selected examples. While the paper provides a theoretical proof claiming a $1-1/e$ approximation guarantee for the proposed algorithm, there are some flaws in the theoretical analysis. Additionally, although the authors present a amount of experimental results, the empirical evaluation is still not comprehensive enough.

**Compliance With Llm Reviewing Policy:**

Affirmed.

**Final Justification:**

I thank the authors for the detailed rebuttal and the effort put into the additional experiments. Given that most of my major concerns have been addressed, I have finalized my score to weak accept.

**Key Questions For Authors:**

Please referring to the Weaknesses.

**Limitations:**

Yes

**Strengths And Weaknesses:**

Strengths:

1. The intuition behind the process-aware mechanism makes practical sense. The dynamic redundancy penalty solves the common issue of over-sampling from high-density regions.

2. Adaptive handling of local neighborhood density. The paper defines local density by using the median distance to the $k$-th nearest neighbors as an adaptive radius ($r$). This approach allows the density calculations to dynamically adjust to different datasets with varying spatial distributions and density scales, ensuring the core sample selection remains stable.

3. The paper is well-written and easy to follow. The motivation is clearly explained, and the limitations of existing geometry-based methods are laid out in a way that naturally leads to the proposed solution.

Weaknesses:

1. Proof for submodularity and monotonicity in the Diversity term

In Eq. (3), dividing the diversity score by the dynamic set size $|S|$ means adding a new sample could actually decrease the overall score. This fundamentally breaks both submodularity and monotonicity. However, the proof in the appendix treats $|S|$ as a constant (a "positive linear transformation"). This disconnect invalidates the $1-1/e$ greedy guarantee, which severely undermines the methodological soundness of the proposed framework.

2. Stability of the $\alpha$ estimation process

Sec 3.4 estimates $\alpha$ by running an unweighted "warm-up" (first $T$ greedy steps) to compute expected marginal gains. But according to Eq. (3) and Eq. (4), the raw Rep score natively dominates Div. This means the unweighted warm-up trajectory will be heavily biased toward high-density regions. Estimating "average" gains from such a highly skewed path seems unstable. The paper is missing an ablation study showing the variance and robustness of this estimation process.

3. Sensitivity to the choice of embedding model

PaCS-ICL is a geometry-based approach, meaning everything—distances $r$, densities $\rho$, and the kernel matrix $K$—is entirely dependent on the underlying text embeddings. Different embedding models yield totally different geometric spaces. It would be helpful to swap in a few different embedding extractors (LLaMa-3 for exmaple) to demonstrate the method's robustness across different representation spaces.

4. The $O(n^2)$ complexity contradicts the "highly scalable" claim (Section 4.6)

Sec 4.6 states that computing local densities and neighborhoods takes $O(n^2)$ time. In real-world NLP, unlabeled pools (e.g., from Wikipedia) can easily hit hundreds of thousands or millions of samples. An $O(n^2)$ pairwise distance computation is unacceptable for both memory and compute at that scale. Did the authors use approximate nearest neighbor (ANN) search like FAISS? This needs to be clarified to back up the scalability claims.

5. Impact of the unlabeled pool size ($|U|$) on ICL performance

Table 5 reports the runtime for different pool sizes (5k, 10k, 30k), but there is no corresponding data on how these sizes actually affect downstream ICL performance. The authors should add experiments showing whether (and how) the quality of the selected coreset changes as the pool size increases across these amounts.

6. Performance under extreme low budgets

The paper primarily evaluates budgets of $m=18$ and $m=100$. However, in real-world few-shot ICL scenarios, we often deal with extremely tight budgets (e.g., $m=3$ or $5$) due to LLM context window limits and attention decay. It would be valuable to include an evaluation at these extreme low budgets ($m \in \{3, 5, 10\}$) to see if PaCS-ICL can still consistently outperform the baselines.

---

> ### Author Rebuttal · Authors · 2026-03-31
>
> ## Q1: About the submodularity
>
> We sincerely thank the reviewer for pointing out the typo in the formula. In our actual implementation (as the code provided in the supplementary material), the $|S|$ term is not included. We confirm that both the code and the experiments use the correct procedure, strictly satisfying submodularity, and the typo does not affect any experimental results or conclusions.
> We have revised the formula and clarified the value ranges.
>
> $\mathrm{Div}(S) = \frac{\log \det(K_S + \epsilon I)}{\log(\lambda_{\max}(K_{\mathcal{U}}) + \epsilon)}$.
>
> The value range of $\mathrm{Div}(S)$  is [0, $|S|$], which is the same as that of $\mathrm{Rep}(S)$.
>
> ---
>
> **Q2: About the computational complexity**
> **A2:** We thank the reviewer for pointing out the computational complexity of the density $\rho$ calculation.
>
>  For moderate-size datasets, our method computes pairwise distances. While the complexity is $O(n^2)$, the computation is highly parallelizable, resulting in very fast runtime, and the density only needs to be computed once.
>
> For large-scale datasets, our method supports approximate nearest neighbor search methods such as FAISS. Specifically, as the radius estimation and density computation in Equations 5–7 only require finding the top-k neighbors or neighbors within radius $r$, approximate search methods can be used. We will clarify this point in the revised manuscript.
>
> We conducted additional experiments to compare runtime with and without FAISS. The FAISS setup used `IndexIVFFlat + range_search`, with `nlist = sqrt(n)` and `nprobe = sqrt(nlist)`.  ` No_Faiss` denotes computing the full pairwise similarity. The GPU experiments used 2 × V100 32GB GPUs.
>
> |Method | Device | SICK (5k) | CQA (10k) | e2eNLG (30k) | MNLI (392k) |
> |--------|------------|-----------|------------|---------------|-------------|
> | No_Faiss    | gpu | 0.6s      | 0.6s       | 3.4s          | 214s        |
> |        | cpu   | 0.3s      | 1.3s       | 15.0s         | 1920s       |
> |  Faiss    | cpu-only    | 0.6s      | 1.8s       | 10.3s         | 397.2s      |
>
> **Conclusion:** On CPU, FAISS is faster than ` No_Faiss ` when the dataset size is ≥30k. On GPU, since FAISS range search does not support GPU, we recommend using ` No_Faiss ` for faster computation. For even larger datasets, FAISS on CPU can be employed.
>
>
> ---
>
> ## Q3: About the ablation of $T$ in probe
>
> We conducted an ablation study on the choice of $T$ (number of probe iterations).
>
> | Dataset   | Method | 0.5m | 1m  | 1.5m | 2m  | 2.5m | 3m  |
> |-----------|--------|------|-----|------|-----|------|-----|
> | MRPC      | PaCS-ICL   | 68.2 | 68.3| 69.1 |69.2 | 69.2 |68.1 |
> | SST-5     |        | 38.2 | 38.2| 38.2 |38.2 | 37.0 |37.0 |
> | CQA       |        | 80.0 | 80.0| 80.0 |80.0 | 80.0 |80.0 |
> | ARC       |        | 81.0 | 81.0| 81.0 |81.0 | 81.0 |81.0 |
> | SICK      |        | 75.3 | 75.3| 75.3 |75.3 | 75.3 |74.2 |
> | MNLI      |        | 73.2 | 73.2| 73.5 |73.5 | 73.5 |73.5 |
> | e2e-NLG   |        | 35.6 | 35.9| 35.4 |35.7 | 35.9 |35.3 |
>
> The method is insensitive to the choice of $T$, and the optimal $T$ typically lies in the range of **1.5m–2.5m**.
>
> ---
>
> ## Q4: About the embedding model
>
> 1. We conducted additional experiments using **LLaMA3** as the embedding model. Specifically, we use **mean pooling over the hidden states of the last layer for each token** to obtain text representations.
>
> | Embedding       | Method   | MRPC | SST-5 | CQA  | ARC  | MNLI | SICK |
> |-----------------|---------|------|-------|------|------|------|------|
> | LLaMA3-hidden   | Infomax | 68.6 | 29.9  | 77.8 | 80.0 | 71.0 | 64.5 |
> |                 | Ours    | 69.4 | 34.7  | 78.0 | 79.9 | 74.8 | 68.9 |
>
> 2. We use **Sentence-BERT** as our embedding model because it is widely adopted in NLP tasks and its effectiveness has been validated in applications such as coreset selection, information retrieval, text clustering, and semantic search. Moreover, the embedding dimension of Sentence-BERT is 768, which is much smaller than LLaMA3's 4096, leading to higher efficiency.
>
>
>
> ---
>
> ## Q5: About the extremely small budgets
>
> We conducted additional experiments to evaluate the performance of our method under **extremely small annotation budgets** ($m = 3, 5, 10$).
>
> | Budget (m) | Method   | MRPC | SST-5 | CQA  | ARC  | MNLI | SICK | e2e-NLG | Avg  |
> |------------|---------|------|-------|------|------|------|------|---------|------|
> | 3          | Infomax | 62.0 | 30.4  | 76.1 | 79.1 | 48.5 | 66.3 | 41.6    | 57.7 |
> | 3          | Ours    | 62.2 | 31.5  | 77.1 | 80.4 | 46.1 | 66.7 | 43.1    | 58.2 |
> | 5          | Infomax | 67.1 | 30.1  | 77.6 | 80.5 | 66.5 | 60.4 | 41.4    | 60.5 |
> | 5          | Ours    | 66.2 | 31.7  | 78.5 | 80.5 | 66.2 | 67.4 | 37.6    | 61.2 |
> | 10         | Infomax | 67.7 | 32.7  | 76.9 | 79.1 | 65.1 | 67.2 | 36.8    | 60.8 |
> | 10         | Ours    | 70.8 | 34.2  | 78.0 | 80.0 | 69.0 | 67.6 | 37.8    | 62.5 |

---

> > ### Author Rebuttal · Reviewer_nLf8 · 2026-04-04
> >
> > I thank the authors for the detailed rebuttal and the effort put into the additional experiments. The response clears up my concern regarding the theoretical soundness. The explanation provided in the rebuttal resolves the previous issues I had with the submodularity proof. However, two important empirical concerns remain unaddressed. First, the authors skipped my question regarding how scaling the unlabeled pool size affects the actual downstream ICL performance, rather than just the runtime. Second, their response to the $\alpha$ estimation stability missed the core point: ablating the number of steps ($T$) does not address the fundamental question of whether the "unweighted warm-up trajectory" will be biased toward high-density regions. Given these unresolved gaps, I have decided to maintain my current score.

---

> > > ### Author Response · Authors · 2026-04-08
> > >
> > > We sincerely thank the reviewer for raising the insightful questions.  We provide the following additional analysis and experiments.
> > >
> > > ### About scaling effectiveness on unlabeled set sizes
> > >
> > > We evaluated sample selection on subsets of varying sizes from MNLI (392k) and CQA (10k), reporting $mean$ and $std$ over 3 random seeds.
> > >
> > > #### CQA
> > > | Budget | Method   | 1k                     | 2k                     | 4k                     | 8k                     | 10k    |
> > > |--------|----------|------------------------|------------------------|------------------------|------------------------|--------|
> > > | 18     | infomax  | 77.9$\pm$0.65            | 78.3$\pm$0.61            | 78.1$\pm$1.21            | 78.0$\pm$0.23            | 76.7   |
> > > | 18     | PaCS-ICL      | 80.3$\pm$0.92            | 80.9$\pm$0.47            | 81.9$\pm$0.72            | 80.8$\pm$0.49            | 80.0   |
> > > | 100    | infomax  | 77.7$\pm$0.32            | 77.5$\pm$0.28            | 77.9$\pm$0.15            | 77.5$\pm$0.41            | 78.1   |
> > > | 100    | PaCS-ICL    | 77.6$\pm$0.05            | 77.9$\pm$0.11            | 78.1$\pm$0.00            | 78.8$\pm$0.85            | 79.4   |
> > >
> > > #### MNLI
> > > | Budget | Method   | 1k                     | 2k                     | 4k                     | 10k                    | 20k                    | 40k                    | 392k   |
> > > |--------|----------|------------------------|------------------------|------------------------|------------------------|------------------------|------------------------|--------|
> > > | 18     | Infomax  | 73.0$\pm$2.13              | 71.6$\pm$2.4               | 70.7$\pm$1.27              | 72.6$\pm$0.1               | 71.1$\pm$0.82              | 70.2$\pm$2.08              | 70.0   |
> > > | 18     | PaCS-ICL      | 72.6$\pm$0.48              | 70.6$\pm$2.30              | 72.2$\pm$2.27              | 72.9$\pm$1.90              | 73.4$\pm$0.94              | 70.9+0.20              | 70.9   |
> > > | 100    | Infomax  | 71.1$\pm$0.50              | 71.3$\pm$1.73              | 71.5$\pm$0.89              | 72.1$\pm$1.01              | 71.8$\pm$0.32              | 71.0+1.68              | 71.4   |
> > > | 100    | PaCS-ICL     | 72.6$\pm$0.61              | 72.7$\pm$0.64              | 72.0$\pm$0.79              | 72.2$\pm$0.53              | 72.4$\pm$0.66              | 72.5$\pm$0.36              | 73.8   |
> > >
> > > For a low budget of 18, performance peaks at a moderate pool size (e.g., 4k for PaCS‑ICL, 2k for Infomax on CQA, and 20k for PaCS‑ICL on MNLI), with further pool expansion bringing no gain. For a higher budget of 100, enlarging the pool consistently improves performance (monotonically on CQA, increasing trend on MNLI).
> > >
> > > ### On the Probe / Warm-up
> > > The probe phase does not repeatedly select the same high-density region because Eq. (7) suppresses repeated selection within radius $r$. Using the median to define $r$ (Eq. (5)) further mitigates the excessive influence of high‑density regions to some extent.
> > >
> > > The $\Delta Rep$ decays exponentially as different high‑density regions become saturated in coverage, while $\Delta Div$ remains stable.  The selection process radiates from high-density regions outward beyond the radius $ r$. Thus, their average gain ratio over a $T$-step probe can reflect their relative scales.
> > >
> > > We clarify that high-density regions are not harmful; they are preferred for their representativeness. The harm lies in the redundancy caused by repeatedly selecting the same region.
> > >
> > > ### Stability analysis of probe estimates
> > >
> > > To assess the stability of the probe process, we inject Gaussian noise $\mathcal{N}(0, \sigma^2)$ with $\sigma = 0.1$ (53% of the original std.)  into the score of each candidate sample at each greedy step to influence the selection trajectory. Over 10 independent runs, we report the  $mean/variance$ of $\frac{\mathbb{E}\Delta Div}{\mathbb{E}\Delta Rep}$ across runs.
> > >
> > > | dataset | T=0.5m | T=m | T=1.5m | T=2m | T=2.5m | T=3m |
> > > |---------|--------|-----|--------|------|--------|------|
> > > | MRPC    | 0.11/0.0001 | 0.13/0.0003 | 0.15/0.0002 | 0.16/0.0002 | 0.18/0.0002 | 0.20/0.0002 |
> > > | SST5    | 0.23/0.0030 | 0.31/0.0020 | 0.38/0.0020 | 0.42/0.0030 | 0.47/0.0030 | 0.51/0.0030 |
> > > | CQA     | 0.17/0.0024 | 0.21/0.0010 | 0.27/0.0074 | 0.31/0.0013 | 0.33/0.0017 | 0.35/0.0014 |
> > > | SICK    | 0.09/0.0003 | 0.10/0.0003 | 0.12/0.0006 | 0.13/0.0002 | 0.14/0.0005 | 0.15/0.0004 |
> > >
> > > As $T$ increases, the mean of $\frac{\mathbb{E}\Delta Div}{\mathbb{E}\Delta Rep}$ increases, indicating that the contribution of $Rep$ decreases. The very small variance shows that the probe estimate is highly stable against noise.
> > >
> > > We set $m < T \ll n$ to focus on "active samples" — those with a high potential of being selected in the final selection process. Analysis of the effect of $T$ on ICL performance shows that $T$ is optimal in $[1.5m,\, 2.5m]$ and insensitive within this range.
> > >
> > > We thank the reviewer for the helpful comments, and we hope this response clarifies our design and reasoning.

---

### Official Review · Reviewer_bWvf · 2026-03-13

**Soundness:** 2
**Presentation:** 3
**Significance:** 2
**Originality:** 3
**Overall Recommendation:** 4
**Confidence:** 3

**Summary:**

The paper studies unsupervised coreset selection for few-shot in-context learning. The proposed method, PaCS-ICL, maximizes a greedy objective, combining diversity and representativeness. To balance these two objectives, the method introduces a coefficient $\alpha$, which is estimated using a budget-aware factor together with a probe sequence of marginal gains. Empirically, the method achieves better average performance, particularly under tighter annotation budgets. The paper also includes several ablation and analysis experiments, including retrieval robustness, radius sensitivity, and runtime evaluation.

**Compliance With Llm Reviewing Policy:**

Affirmed.

**Final Justification:**

Thank you for the detailed rebuttal and for adding the new discussion and comparative experiments. The response makes the paper’s scope and positioning clearer, and the additional evidence strengthens the empirical case, so I have increased my score accordingly. That said, I would still encourage the authors to include more baselines from the query-dependent demonstration selection literature to better clarify the practical scope and performance ceiling of the proposed approach.

**Key Questions For Authors:**

- Can the authors compare against simpler geometric baseline like facility location?

**Limitations:**

A main limitation of this work is that its effectiveness depends heavily on the geometry of the pretrained embedding space. In addition, scalability remains a concern, as the method still incurs relatively high computational cost when the unlabeled pool or selection budget becomes large.

**Strengths And Weaknesses:**

### Strengths

- The problem is practical. Selecting a small unlabeled subset for later annotation is a useful formulation for budgeted ICL.
- The method is intuitively sensible: density-aware representativeness plus redundancy control is a reasonable way to avoid both peripheral outliers and over-concentration in dense regions.
- The empirical section is fairly solid, with experiments conducted on multiple datasets and backbone models, along with a substantial number of ablations and analysis studies.

### Weaknesses

- PaCS-ICL selects a single global coreset from the unlabeled pool without conditioning on individual input queries, even though the final prompt construction is query-specific. Moreover, this query-specific stage is implemented only through a simple similarity-based retrieval step, rather than being jointly optimized with the coreset selection itself. As a result, the method is better characterized as a global pool construction approach than a direct solution to query-specific demonstration selection, which may limit its ultimate performance ceiling.
- While the paper compare relevant coreset baselines, the final ICL prompts are still constructed through downstream retrieval, so it would strengthen the empirical case to compare against stronger query-aware selectors as practical baselines, such as RDES[1], GenICL[2] or ICL-GRAD [3]. Even if these methods are not perfectly aligned with the unlabeled coreset setting, the lack of simple adaptations or at least a discussion of them makes the empirical evaluation feel somewhat narrow.
- For the diversity term, the appendix argues that $\mathrm{Div}(S)$ is submodular because it is a positive linear transformation of $\log\det(K_S+\epsilon I)$. However, the normalization includes a division by $|S|$, which is set-dependent rather than a constant scaling, so the current proof does not actually establish submodularity of the normalized objective in Eq.(3). For the representativeness term, the appendix first informally states the incremental gain as $\rho_t\gamma_t$, but the actual marginal gain expression later includes additional terms over previously selected items, and those terms are non-positive. As a result, the monotonicity of $\mathrm{Rep}(S)$, are not clearly established by the current derivation.
- The paper reports an overall complexity of $O(n^2 + nm^2)$, with the $O(n^2)$ neighborhood computation identified as the dominant step. In the runtime table, PaCS-ICL is much slower than InfoMax. While this does not make the method unusable, it weakens the claim that the approach is broadly scalable, especially if one wishes to move beyond the current few-shot budgets in the long-context scenarios.

[1] Demonstration Selection for In-Context Learning via Reinforcement Learning

[2] Learning to Select In-Context Demonstration Preferred by Large Language Model

[3] Linear-Time Demonstration Selection for In-Context Learning via Gradient Estimation

---

> ### Author Rebuttal · Authors · 2026-03-31
>
> **Q1: About the submodularity**
> **A1:** We sincerely thank the reviewer for pointing out the typo in the formula. In our actual implementation (as the code submitted in the supplementary material), the marginal gain of $\text{Rep}(S)$ is computed as
>
> $$
> \Delta \text{Rep}\left(x \mid S\right) = \rho_x \cdot \gamma_x(S)
> $$
>
> i.e., it only accounts for the contribution of the newly added element, and does not consider any  adjustment from elements already in the set.
>
> Formally, for any sets $A \subseteq B \subseteq U$, we have
>
> $$
> \gamma_x(B) \le \gamma_x(A)
> $$
> which ensures that our implementation strictly satisfies the diminishing returns property of submodularity.
>
> And the $|S|$ term is not included in our implementation. We have revised the formula and clarified the value ranges.
>
> $\mathrm{Div}(S) = \frac{\log \det(K_S + \epsilon I)}{\log(\lambda_{\max}(K_{\mathcal{U}}) + \epsilon)}$.
>
> The value range of $\mathrm{Div}(S)$  is [0, $|S|$], which is the same as that of $\mathrm{Rep}(S)$.
>
> We confirm that both the code and the experiments use this correct procedure, and therefore the typo does not affect any experimental results or conclusions. We will correct the formula in the revised manuscript.
>
>
> **Q2: About the computational complexity**
> **A2:** We thank the reviewer for pointing out the computational complexity of the density $\rho$ calculation.
>
>  For moderate-size datasets, our method computes pairwise distances. While the complexity is $O(n^2)$, the computation is highly parallelizable, resulting in very fast runtime, and the density only needs to be computed once.
>
> For large-scale datasets, our method supports approximate nearest neighbor search methods such as FAISS. Specifically, as the radius estimation and density computation in Equations 5–7 only require finding the top-k neighbors or neighbors within radius $r$, approximate search methods can be used. We will clarify this point in the revised manuscript.
>
> We conducted additional experiments to compare runtime with and without FAISS. The FAISS setup used `IndexIVFFlat + range_search`, with `nlist = sqrt(n)` and `nprobe = sqrt(nlist)`.  ` No_Faiss` denotes computing the full pairwise similarity. The GPU experiments used 2 × V100 32GB GPUs.
>
> |Method | Device | SICK (5k) | CQA (10k) | e2eNLG (30k) | MNLI (392k) |
> |--------|------------|-----------|------------|---------------|-------------|
> | No_Faiss    | gpu | 0.6s      | 0.6s       | 3.4s          | 214s        |
> |        | cpu   | 0.3s      | 1.3s       | 15.0s         | 1920s       |
> |  Faiss    | cpu-only    | 0.6s      | 1.8s       | 10.3s         | 397.2s      |
>
> **Conclusion:** On CPU, FAISS is faster than ` No_Faiss ` when the dataset size is ≥30k. On GPU, since FAISS range search does not support GPU, we recommend using ` No_Faiss ` for faster computation. For even larger datasets, FAISS on CPU can be employed.
>
> **Q3: About the ICL**
>
>
>
> **A3:** In this work, **ICL** primarily focuses on the **select-then-annotation[1]** workflow under scenarios with a small annotation budget. Compared with coreset for model training, this setting imposes stricter requirements on the quality of labeled data, which should capture global features of the dataset. These features are query-independent and can thus generalize to **unseen queries**. Our method is specifically designed to achieve this **query-independent global selection objective**, aligning with the standard formulation of unsupervised coreset selection. Experimental results (Table 3) demonstrate that our method achieves robust performance across various retrieval strategies.
>
> [1] Su H, Kasai J, Wu C H, et al. Selective annotation makes language models better few-shot learners. In ICLR, 2023.
>
> **Q4: About comparison with Facility Location**
>
> **A4:** We compare our method with the **Facility Location** greedy selection baseline. In this baseline, samples are selected to maximize coverage of the embedding space. Specifically, it iteratively selects the point that maximizes the incremental gain in coverage, defined as the increase in the maximum similarity between each point and the current selected set.
>
> | m | Method | MRPC | SST-5 | CQA | ARC | MNLI | SICK |
> |-----|-----------------|------|-------|------|------|------|------|
> | 18 | Facility Location | 53.6 | 33.9 | 76.7 | 76.3 | 49.1 | 54.3 |
> | 18 | Ours | 69.3 | 38.2 | 80.0 | 81.0 | 73.5 | 75.3 |
> | 100 | Facility Location | 53.7 | 33.8 | 76.7 | 76.2 | 49.1 | 54.3 |
> | 100 | Ours | 68.7 | 37.6 | 79.4 | 81.6 | 72.4 | 73.4 |

---

> > ### Author Rebuttal · Reviewer_bWvf · 2026-04-02
> >
> > Thank you for the detailed clarifications in the rebuttal. I appreciate the additional explanation regarding both the implementation and the intended problem setting. On the formula issue, I would still view the original presentation as more than a minor typo. The rebuttal helps clarify what was actually implemented, but I still think the paper should explain this point more carefully in the revised version.
> >
> > Regarding scope, I understand from the rebuttal that the paper is intentionally positioned in the select-then-annotation setting, which is consistent with prior work such as IDEAL. I think this is a reasonable setting, so my concern is not that the paper addresses the wrong problem. Rather, this setting appears relatively narrow, and its current significance is not yet fully established. In particular, the more recent baselines considered in the paper, such as RDSS and InfoMax, are closer to general coreset selection than to the ICL-specific problem. At the same time, much of the recent ICL literature has increasingly focused on query-dependent retrieval and inference-time example selection, as reflected in retrieval-based ICL surveys and newer methods such as RDES and GenICL, which are evaluated on similar types of downstream tasks. I therefore think the paper would benefit from a clearer discussion/direct comparison of the motivation, advantages, limitations, and intended use cases of this setting. Otherwise, under the current focus of the ICL literature, the contribution may be interpreted more narrowly.
> >
> > As for generalizability, this issue is also explicitly considered and empirically validated in many query-dependent methods. Without further empirical evidence or more direct comparison, I would be inclined to maintain my score.

---

> > > ### Author Response · Authors · 2026-04-07
> > >
> > > Thank you for your thoughtful consideration. We agree that discussing query-based approaches is a valuable addition. We add both a discussion and comparative experiments against query-based ICL methods.
> > >
> > > ### Discussion
> > >
> > > The strength of query-based ICL lies in its ability to handle extremely long-tailed query distributions, as it can tailor the most relevant local context for each long-tail query. The limitations are that it relies on sufficient labeled data to characterize the query distribution, and its computational cost is often substantial. For instance, the baselines involve the LLM itself in the sample selection process — whether through model gradient information [1] or via training [2][3]. Therefore, query-based methods are well-suited for scenarios where a reasonable amount of labeled data is already available, enabling fine-grained query-aware demonstration selection, especially when the query distribution is highly long-tailed.
> > >
> > > The applicability of *unsupervised coreset selection* (UCS) complements that of query-based methods. UCS selects samples without relying on the label information, is independent of the LLM, and incurs relatively low computational cost. It is well-suited for cold-start scenarios and rapid transfer to new data or tasks — such as streaming data from news or social media, where data emerges continuously and annotation inevitably lags. UCS is also suited to static, low-resource settings, such as expert-dependent domain tasks with tight annotation budgets. Its limitation is that for certain highly long-tailed or highly heterogeneous query data, global selection becomes more challenging and may not be as precise as dynamic retrieval-based methods.
> > >
> > > ### Comparative Experiments
> > >
> > > We added comparative experiments against GradeRE [1]. It leverages the gradient information of LLMs to estimate the influence of candidate demos on the query set, thereby selecting the most effective subset of demos for the query set.
> > >
> > > **Setup:** We evaluate on two QA tasks (CQA, ARC) and two classification tasks (MRPC, SST5), using Llama-3-8B as the base model.
> > >
> > > For GradeRE [1], we follow its official implementation and adopt the fully supervised setting, where we assume access to all labeled data from the original training set. Specifically, we randomly sample 500 labeled samples from the original training set to serve as the query training set. The remaining labeled samples in the original training set are used as candidates. The number of candidate subsets is set to 500, and the number of anchor sets is set to 1.
> > >
> > > We experiment with two demonstration counts: 5 (following their setting) and 18 (our setting). The sizes of the candidate subset and the anchor set match the demonstration count (i.e., 5 or 18, respectively).
> > >
> > > **#demos=18**
> > >
> > > | budget | methods | MRPC | SST5 | CQA | ARC |
> > > |--------|---------|------|------|-----|-----|
> > > | supervised | GradeRE | 65.7 | 33.4 | 78.6 | 79.9 |
> > > | 18 | PaCS-ICL | 69.3 | 38.2 | 80.0 | 81.0 |
> > > | 100 | PaCS-ICL | 68.7 | 37.6 | 79.4 | 81.6 |
> > >
> > > **#demos=5**
> > >
> > > | budget | methods | MRPC | SST5 | CQA | ARC |
> > > |--------|---------|------|------|-----|-----|
> > > | supervised | GradeRE | 58.5 | 30.7 | 77.0 | 79.4 |
> > > | 18 | PaCS-ICL | 60.3 | 31.5 | 78.3 | 80.4 |
> > > | 100 | PaCS-ICL | 59.3 | 30.6 | 78.0 | 80.0 |
> > >
> > > We note that GradeRE [1] also provides a solution for low-label settings (Appendix B.4), where they report that using pseudo-labels or random labels achieves the same accuracy as using ground-truth labels. Therefore, the above GradeRE results can be reasonably approximated as the performance under a labeling budget of approximately 500 examples (for the query set plus anchor set). Even under this approximation, our method still shows consistent advantages, suggesting that when the data is moderately sized and not extremely long-tailed or highly heterogeneous,  a small labeling budget may suffice to achieve strong performance.
> > >
> > > We thank the reviewer again for the insightful suggestions. We have added the corresponding experiments and hope that they address your concerns.
> > >
> > > ---
> > >
> > > [1] Linear-Time Demonstration Selection for In-Context Learning via Gradient Estimation
> > > [2] Learning to Select In-Context Demonstration Preferred by Large Language Model
> > > [3] Demonstration Selection for In-Context Learning via Reinforcement Learning

---

### Official Review · Reviewer_hpqH · 2026-03-13

**Soundness:** 2
**Presentation:** 3
**Significance:** 3
**Originality:** 2
**Overall Recommendation:** 4
**Confidence:** 4

**Summary:**

The paper proposes PaCS-ICL, an unsupervised coreset selection framework for In-Context Learning (ICL). The authors argue that existing geometry-based selection methods struggle to balance representativeness (density) and diversity, often leading to skewed or redundant prompt sets. To address this, PaCS-ICL formulates coreset selection as a submodular maximization problem. The objective function linearly combines a normalized Determinantal Point Process (DPP) term for diversity and a density-aware term with an exponential decay factor for representativeness. To balance these terms dynamically across different budgets, the authors introduce a heuristic scaling factor $\alpha$. The coreset is then constructed using a standard greedy algorithm. The method is evaluated on 7 NLP datasets across two LLM backbones (LLaMA2-7B and LLaMA3-8B), demonstrating consistent improvements over baselines like IDEAL and InfoMax under tight annotation budgets (e.g., $m=18$ and $100$).

**Compliance With Llm Reviewing Policy:**

Affirmed.

**Final Justification:**

I will finalize my score at Weak Accept. I expect the authors to rigorously correct the mathematical formulations in the main text and appendix to exactly match their code in the camera-ready version.

**Key Questions For Authors:**

See weaknesses.

**Limitations:**

yes

**Strengths And Weaknesses:**

Strengths:
1. The authors provide a robust empirical evaluation across a diverse set of 7 NLP tasks (ranging from NLI to Generation) using two different modern LLM backbones. The performance gains, particularly in the extreme few-shot setting ($m=18$), are solid and convincing.
2. The analysis of the marginal gain contributions in Section 4.5 and Figure 4 provides excellent transparency into how the algorithm negotiates the trade-off between diversity and representativeness during the greedy selection process.
Weaknesses:
1. The authors heavily rely on the claim that their objective function is monotone submodular to invoke the standard $(1-1/e)$ greedy approximation guarantee. However, the $Rep(S)$ function defined in Eq. 4 is NOT monotonically non-decreasing. In Appendix A.1.3, the authors explicitly derive the marginal gain $\Delta Rep(x|S) = \rho_x \cdot \gamma_x(S) + \sum_{i \in S} \rho_i \cdot (\gamma_i(S \cup \{x\}) - \gamma_i(S))$. The authors correctly note that $\gamma_i(S \cup \{x\}) - \gamma_i(S) \le 0$. This means the second term is strictly negative if the new element $x$ falls within the radius $r$ of existing elements. If the positive gain $\rho_x \cdot \gamma_x(S)$ is small, but the negative penalty applied to existing elements is large, the overall marginal gain $\Delta Rep(x|S)$ becomes negative. A function that can decrease when an element is added is not monotone. Consequently, the classic $(1-1/e)$ bound for the greedy algorithm does not apply here (maximizing non-monotone submodular functions requires entirely different algorithms, such as Double Greedy). This invalidates the core theoretical guarantee of the paper.
2. The paper claims to propose an "adaptive estimation" for the balance factor $\alpha$ (Eq. 8). However, the budget-aware term $\phi(m)$ in Eq. 9 is heavily hand-crafted and clearly overfitted to the experimental settings. The authors define $\phi(m) = \exp(clip(\frac{\lceil n/M \rceil - m}{M}, -1, 1))$ and arbitrarily set the "reference maximum budget" $M=100$. This highly specific, non-generalizable formula appears engineered specifically to yield good results for their chosen budgets of 18 and 100. It lacks any rigorous mathematical derivation and undermines the "principled" nature of the framework.
3. The authors claim the method is robust and scalable, yet step (i) of their algorithm requires computing an $O(n^2)$ pairwise distance matrix for local density estimation. For modern large-scale datasets, $O(n^2)$ complexity is prohibitive. This scalability bottleneck is tacitly admitted in Appendix A.2, where the authors reveal they had to randomly downsample the MNLI training set from 392k to just 5,000 samples to make coreset selection feasible. A coreset selection method that cannot process a standard NLI dataset without aggressive random downsampling is not practically scalable.
4. Rebranding the standard marginal gain evaluation of the greedy algorithm as "Process-Aware Coreset Selection" is borderline math-washing. The fact that the value of adding a new sample depends on the previously selected samples is the fundamental definition of submodularity and the greedy algorithm itself, which has existed for decades.

---

> ### Author Rebuttal · Authors · 2026-03-31
>
> **Q1: About the submodularity**
> **A1:** We sincerely thank the reviewer for pointing out the typo in the formula in Appendix A.1.3. In our actual implementation (as the code submitted in the supplementary material), the marginal gain of $\text{Rep}(S)$ is computed as
>
> $$
> \Delta \text{Rep}\left(x \mid S\right) = \rho_x \cdot \gamma_x(S)
> $$
>
> i.e., it only accounts for the contribution of the newly added element, and does not consider any negative adjustment from elements already in the set.
>
> Formally, for any sets $A \subseteq B \subseteq U$, we have
>
> $$
> \gamma_x(B) \le \gamma_x(A)
> $$
> which ensures that our implementation strictly satisfies the diminishing returns property of submodularity.
>
> We confirm that both the code and the experiments use this correct procedure, and therefore the typo does not affect any experimental results or conclusions. We will correct the formula in the revised manuscript.
>
> ---
> **Q2: About the computational complexity**
> **A2:** We thank the reviewer for pointing out the computational complexity of the density $\rho$ calculation.
>
>  For moderate-size datasets, our method computes pairwise distances. While the complexity is $O(n^2)$, the computation is highly parallelizable, resulting in very fast runtime, and the density only needs to be computed once.
>
> For large-scale datasets, our method supports approximate nearest neighbor search methods such as FAISS. Specifically, as the radius estimation and density computation in Equations 5–7 only require finding the top-k neighbors or neighbors within radius $r$, approximate search methods can be used. We will clarify this point in the revised manuscript.
>
> We conducted additional experiments to compare runtime with and without FAISS. The FAISS setup used `IndexIVFFlat + range_search`, with `nlist = sqrt(n)` and `nprobe = sqrt(nlist)`.  ` No_Faiss` denotes computing the full pairwise similarity. The GPU experiments used 2 × V100 32GB GPUs.
>
> |Method | Device | SICK (5k) | CQA (10k) | e2eNLG (30k) | MNLI (392k) |
> |--------|------------|-----------|------------|---------------|-------------|
> | No_Faiss    | gpu | 0.6s      | 0.6s       | 3.4s          | 214s        |
> |        | cpu   | 0.3s      | 1.3s       | 15.0s         | 1920s       |
> |  Faiss    | cpu-only    | 0.6s      | 1.8s       | 10.3s         | 397.2s      |
>
> **Conclusion:** On CPU, FAISS is faster than ` No_Faiss ` when the dataset size is ≥30k. On GPU, since FAISS range search does not support GPU, we recommend using ` No_Faiss ` for faster computation. For even larger datasets, FAISS on CPU can be employed.
>
>
>
>
>
> ---
>
> **Q3: About the budget-adaptive factor $\phi(m)$**
>
> **A3:**
> The design of $\phi(m)$ is guided by an intuition that the weight for representativeness should adapt to both the annotation budget $m$ and the data scale $n$. A smaller $m$ increases the need for each selected sample to be highly representative, while a larger $n$ generally demands stronger coverage. However, to prevent overemphasizing representativeness in large redundant datasets, we introduce a scaling factor.
>
> We incorporate a reference maximum budget $M$ to normalize the dataset scale. The term $n/M$ is empirical but reasonable, which maps the raw dataset size to the comparable scale as typical small-budget ICL scenarios (tens to around 100).
> $\lceil n/M \rceil - m$ intuitively measures how far the current budget $m$ is from the inherent complexity of covering the dataset.
> The choice of $M=100$ is consistent with prior work [1][2]. It also reflects the observation that the number of useful demonstrations per class tends to saturate [3], making a moderate $M$ is sufficient to balance efficiency and performance.
>
> The effectiveness of this design is not limited to the two settings $m = 18$ and $m = 100$. As shown in Appendix Table 7, we evaluated PaCS-ICL across a continuous range of budgets ($m = 30, 40, \dots, 90$). This demonstrates that $\phi(m)$ is generally effective.
>
>
> [1] Su H, Kasai J, Wu C H, et al. Selective annotation makes language models better few-shot learners. In ICLR, 2023.
>
> [2] Zhang S, Xia X, Wang Z, et al. Ideal: Influence-driven selective annotations empower in-context learners in large language models. In ICLR, 2024.
>
> [3] Li Z, Xu Z, Han L, et al. Implicit in-context learning. In ICLR, 2025.

---

> > ### Author Rebuttal · Reviewer_hpqH · 2026-04-04
> >
> > The authors have successfully defended the empirical utility of their method and patched the severe theoretical and scalability holes identified in my initial review. The framework is practically effective and the code aligns with standard submodular maximization.
> >
> > However, considering that the theoretical elegance is somewhat compromised by the "typo" correction (which simplifies the redundancy mechanism) and the heavy reliance on empirical heuristics for balancing the objective, the overall contribution leans firmly toward solid systems engineering rather than a groundbreaking algorithmic innovation. Therefore, I will finalize my score at Weak Accept. I expect the authors to rigorously correct the mathematical formulations in the main text and appendix to exactly match their code in the camera-ready version

---

> > > ### Author Response · Authors · 2026-04-07
> > >
> > > We sincerely appreciate your valuable suggestions and your positive recognition of the practical effectiveness of our method. Your feedback has been very helpful in revising our manuscript, and we will ensure that the mathematical expressions and code are fully consistent.

---

### Decision · Program_Chairs · 2026-04-30

**Decision:**

Accept (regular)

**Comment:**

This paper studies unsupervised coreset selection for few-shot in-context learning. The proposed method selects a small subset of unlabeled examples by balancing diversity and representativeness under a fixed annotation budget.

Reviewers agreed that the paper studies a meaningful problem, and several appreciated their motivation and the generally strong empirical results across multiple datasets and two LLM backbones. The rebuttal also strengthened the paper by adding several experiments and discussion of the intended select-then-annotation setting.

At the same time, some limitations remain. Multiple reviewers raised concerns about the theoretical presentation such as the submodularity and monotonicity arguments. The rebuttal clarified that the main issue was a mismatch between the written formula and the implemented algorithm, but the final version should revise the mathematical presentation carefully. Reviewers also noted that the method is best understood as a query-independent global pool construction approach in the select-then-annotation setting, rather than a direct solution to query-specific demonstration selection, and that this scope should be stated more explicitly.

Overall, I find that the paper makes a useful contribution in a meaningful but somewhat scoped setting. The final version should better align the theory with the implemented objective and clarify the intended applicability of the method. I lean toward acceptance.